# KLF7 is a general inducer of human pluripotency

Mattia Arboit[1,5], Irene Zorzan [ID][2,4,5], Eleonora Pensabene [ID][1], Marco Pellegrini [ID][2], Federica Bertelli[1], Giorgia Panebianco [ID][1], Davide Benvegnù[1], Giada Rossignoli [ID][1], Paolo Martini [ID][3], Gianluca Amadei[1], Elena Carbognin [ID][1✉] & Graziano Martello [ID][1✉]

## Abstract

Pluripotency is the capacity to generate all somatic cells and the germ line and is governed by a self-reinforcing network of transcription factors. The forced expression of only some of these factors enables the reprogramming of somatic cells to pluripotency. In murine cells, several Krüppel-like factors (KLFs) have been identified as stabilisers and inducers of pluripotency. Human somatic cells are routinely reprogrammed by expression of KLF4 in combination with OCT4, SOX2 and cMYC (OSKM). An extensive transcriptome analysis revealed, however, that KLF4 is barely expressed in conventional human pluripotent stem cells (PSCs). Here we show that KLF7 is robustly expressed in conventional human PSCs and it allows transcription factor-mediated somatic reprogramming replacing KLF4. Moreover, KLF7 is highly expressed in naive PSCs and its forced expression in conventional hPSCs induces upregulation of naive markers and results in efficient chemical resetting to naive PSCs. KLF7 CRISPRi-mediated silencing, while not affecting maintenance of conventional/primed PSCs, greatly reduces the efficiency of chemical resetting. Our data indicate that KLF7 is a general human pluripotency factor and an inducer of pluripotency.

**Keywords** Human Pluripotency; Resetting; Reprogramming; Krüppel-like Factors; KLFs
**Subject Categories** Chromatin, Transcription & Genomics; Signal Transduction; Stem Cells & Regenerative Medicine

## Introduction

Pluripotent stem cells have been originally derived by culturing epiblast cells of early mouse embryos as mESCs (Evans and Kaufman, 1981; Martin, 1981). The molecular characterisation of these cells allowed the identification of specific pluripotency factors which enabled the induction of pluripotency from somatic cells by transcription factor-mediated reprogramming, and the derivation of induced Pluripotent Stem cells (iPSCs) (Takahashi and Yamanaka, 2006).

To generate iPSCs, the forced expression of a cocktail of four factors (OCT4, SOX2, KLF4 and cMYC) (OSKM) was firstly employed for both murine and human somatic reprogramming (Takahashi et al, 2007; Takahashi and Yamanaka, 2006), despite the species-specific differences.

Human conventional PSCs, derived from human blastocysts as human embryonic stem cells (hESCs) (Thomson et al, 2011) or generated by reprogramming of somatic cells (Yu et al, 2007) as iPSCs, resemble a developmental stage called primed pluripotency corresponding to the epiblast of the post-implantation blastocyst (Davidson et al, 2015; Nakamura et al, 2016). Recently, human naive pluripotent cells, in a more primitive state resembling the pre-implantation embryo, have been derived directly from the human pre-implantation embryo (Guo et al, 2016), by reprogramming of fibroblasts (Giulitti et al, 2019; Kilens et al, 2018; Liu et al, 2017; Wang et al, 2018), by transgene-mediated resetting of human conventional PSCs (Takashima et al, 2014; Theunissen et al, 2014) and by chemical resetting (Bayerl et al, 2021; Guo et al, 2017; Theunissen et al, 2014).

In the murine system, several Krüppel-like factors (KLF2/4/5) are important for the maintenance of pluripotency in naive cells and are absent in primed EpiSCs (Yamane et al, 2018). Indeed, these Krüppel-like factors have been used to generate naive iPS cells from mouse fibroblasts, in combination with OSM, and are all able to reset primed EpiSCs to naive pluripotency (Dunn et al, 2019; Nakagawa et al, 2008; Jeon et al, 2016).

In human naive cells, KLF4, KLF5 and KLF17 are highly expressed (Blakeley et al, 2015; Boroviak et al, 2015; Ai et al, 2022), but their expression in conventional/primed PSCs remains unexplored. Analysis of human pre- and post-implantation epiblast cells indicates that these Krüppel-like factors are expressed specifically in the naive state (Xiang et al, 2020). Indeed, KLF17 and KLF4 have been shown to be powerful inducers of naive pluripotency in vitro (Lea et al, 2021; Guo et al, 2009), and KLF4 is also routinely used to generate conventional human iPSCs (Takahashi et al, 2007).

We performed a thorough transcriptional analysis of conventional PSCs cultured under different conditions and consistently failed to detect robust KLF4 expression. In contrast, a member of

[1]Department of Biology, University of Padua, Padua, Italy. [2]Department of Molecular Medicine, Medical School, University of Padua, Padua, Italy. [3]Department of Molecular and Translational Medicine, University of Brescia, Brescia, Italy. [4]Present address: Epigenetics Programme, Babraham Institute, Cambridge, UK. [5]These authors contributed equally: Mattia Arboit, Irene Zorzan. ✉E-mail: elena.carbognin@unipd.it; graziano.martello@unipd.it

the KLF family, named KLF7, is robustly expressed in conventional PSCs and supports pluripotency downstream of TGF-beta (Zorzan et al, 2020). KLF7 is also highly expressed in human naive PSCs. Expression of KLF7 alongside reprogramming factors OCT4, SOX2 and cMYC, enables robust human somatic reprogramming. Forced expression of KLF7 in conventional PSCs maximises chemical resetting to naive pluripotency, while its downregulation severely reduces resetting efficiency. We thus conclude that KLF7 is a general inducer of human pluripotency.

## Results and discussion

### KLF7 enables the induction of pluripotency

Induction of pluripotency in human somatic cells can be achieved via the expression of different combinations of the factors OCT4, SOX2, KLF4, cMYC, NANOG, LIN28A (OSKM and OSNL), which leads to the generation of induced pluripotent stem cells (iPSCs) (Takahashi et al, 2007; Yu et al, 2007). We verified the absolute expression of such reprogramming factors in several human PSC and iPSC lines by re-analysing available RNAseq data (Choi et al, 2015; Dong et al, 2020; Giulitti et al, 2019; Jang et al, 2022; Liu et al, 2017; Theunissen et al, 2016; Wei et al, 2021; Zorzan et al, 2020, 2023) and observed that expression of the pluripotency factor KLF4 was barely detectable (Fig. 1A). This prompted us to measure the expression of other Krüppel-like factors previously involved in pluripotency maintenance or induction in either human or murine cells (KLF2/4/5/7/17) and noticed that KLF7 is the most represented amongst the others, at levels comparable to other transcription factors functionally involved in the maintenance and induction of pluripotency, such as PRDM14 (Chia et al, 2010) and ZNF398 (Zorzan et al, 2020) (Fig. 1B). The other KLFs analysed were all expressed at low levels, comparable to markers of early differentiation such as SOX1 and T (also known as BRACHYURY or T). Of note, we systematically failed to detect expression of KLF4, regardless of the origin of PSCs analysed, either embryo-derived (ESCs) or via reprogramming of somatic cells (iPSCs) (Fig. EV1A).

Since KLF7 sustains pluripotency in conventional PSCs (Zorzan et al, 2020), we hypothesised that expression of KLF7 could perform a similar function as KLF4 and, together with OCT4, SOX2 and cMYC (OSK7M), could enable reprogramming of primary somatic cells.

To test this hypothesis, we delivered modified messenger RNAs (mmRNAs)(Warren et al, 2010; Zorzan et al, 2022) encoding for OSK7M to human BJ fibroblasts, using microfluidics which have been reported to lead to rapid and efficient generation of primed iPSCs (Luni et al, 2016). We also used OSKM and OSNL (Takahashi et al, 2007; Yu et al, 2007) as positive controls. By day 14, iPSC colonies with morphology of primed pluripotent stem cells were detected (dashed circles in Fig. EV1B). Immunofluorescence (IF) for pluripotency markers OCT4 and NANOG confirmed acquisition of pluripotency (Fig. 1C).

All three reprogramming cocktails used generated iPSC colonies, although OSNL was less robust, less efficient and formed smaller colonies (Figs. 1C–E and EV1B).

We then expanded the newly generated iPSCs to obtain stable lines. Colonies derived from OSKM and OSK7M cocktails could be readily propagated for 20 passages while primary iPSC colonies generated with OSNL could not stabilise in culture, therefore were not further characterised (Fig. EV1C). Gene expression analysis (Fig. EV1D,E) revealed that OSK7M iPSCs, up to 20 passages, express pluripotency markers at levels comparable to conventional human Embryonic Stem cells (hESCs - H9) and conventional iPSCs generated with OSKM from keratinocytes (kiPSCs).

These results indicate that expression of KLF7, together with OCT4, SOX2 and cMYC, enables efficient and robust human somatic cell reprogramming.

In order to further characterise these newly established iPSC lines, we performed transcriptome analyses. Unsupervised clustering showed that the transcriptional profile is clearly distinct from that of fibroblasts and comparable to hESCs (H9 cells) (Fig. 1F). Moreover, analysis of a large panel of pluripotency and fibroblast-specific genes (Fig. 1G) further corroborated the evidence that stable iPSC lines obtained from reprogramming with OSK7M are transcriptionally comparable to embryo-derived PSCs.

To assess the quality of OSK7M iPSCs and their capacity to differentiate towards the three germ layers, we performed embryoid body (EB) differentiation (Fig. EV2A). After 15 days, we could not detect differences in the size and shape of EBs (Fig. EV2B). Molecularly, we observed a reduction of pluripotency markers and comparable levels of lineage markers in all iPSC lines analysed, confirming that they are pluripotent and retain multi-lineage potential. Similar results were obtained with iPS cells expanded for 7, 15 or 20 passages (Figs. 1H and EV2C,D). Taken together, these results clearly show that reprogramming with OSK7M generated bona fide human iPSCs.

### KLF7 improves the efficiency of chemical resetting

We analysed the transcriptome of naive and conventional hPSCs and human fibroblasts, and observed that KLF7 is expressed in naive hPSCs and in primed hPSCs at comparable levels, like other known general pluripotency markers (e.g. NANOG, POU5F1, SALL4, LIN28B) (Fig. 2A,B).

In murine PSCs, Nanog and Oct4 are pluripotency factors expressed both at naive and primed states. Their forced expression, in combination with medium supporting naive pluripotency, has been shown to efficiently reset primed murine PSCs to the naive state (Radzisheuskaya et al, 2013; Theunissen et al, 2011).

Human naive PSCs can be obtained in vitro by chemical resetting of conventional hPSCs via transient inhibition of histone deacetylase (Guo et al, 2017), although with low efficiency. Given that KLF7 is expressed in both naive and conventional hPSCs, we analysed its expression during chemical resetting of conventional PSCs to naive PSCs. After 3 days, we detected a peak in KLF7 expression (Fig. 2C). Our observation was also supported by RNA sequencing data of chemical resetting (Zorzan et al, 2023) (Fig. 2D).

KLF7 expression transiently increases during resetting, so we hypothesised that KLF7 forced expression might enhance chemical resetting of human PSCs. We generated conventional iPSCs stably expressing either KLF7 (KLF7-iPSCs) or an empty vector (EMPTY-iPSCs) and then subjected them to the chemical resetting protocol (Fig. 3A).

In EMPTY-iPSCs, resetting gave rise to a mixed population, as previously reported (Guo et al, 2017; Zorzan et al, 2023), while overexpression of KLF7 resulted in a morphologically homogeneous population of naive colonies (Fig. 3B).

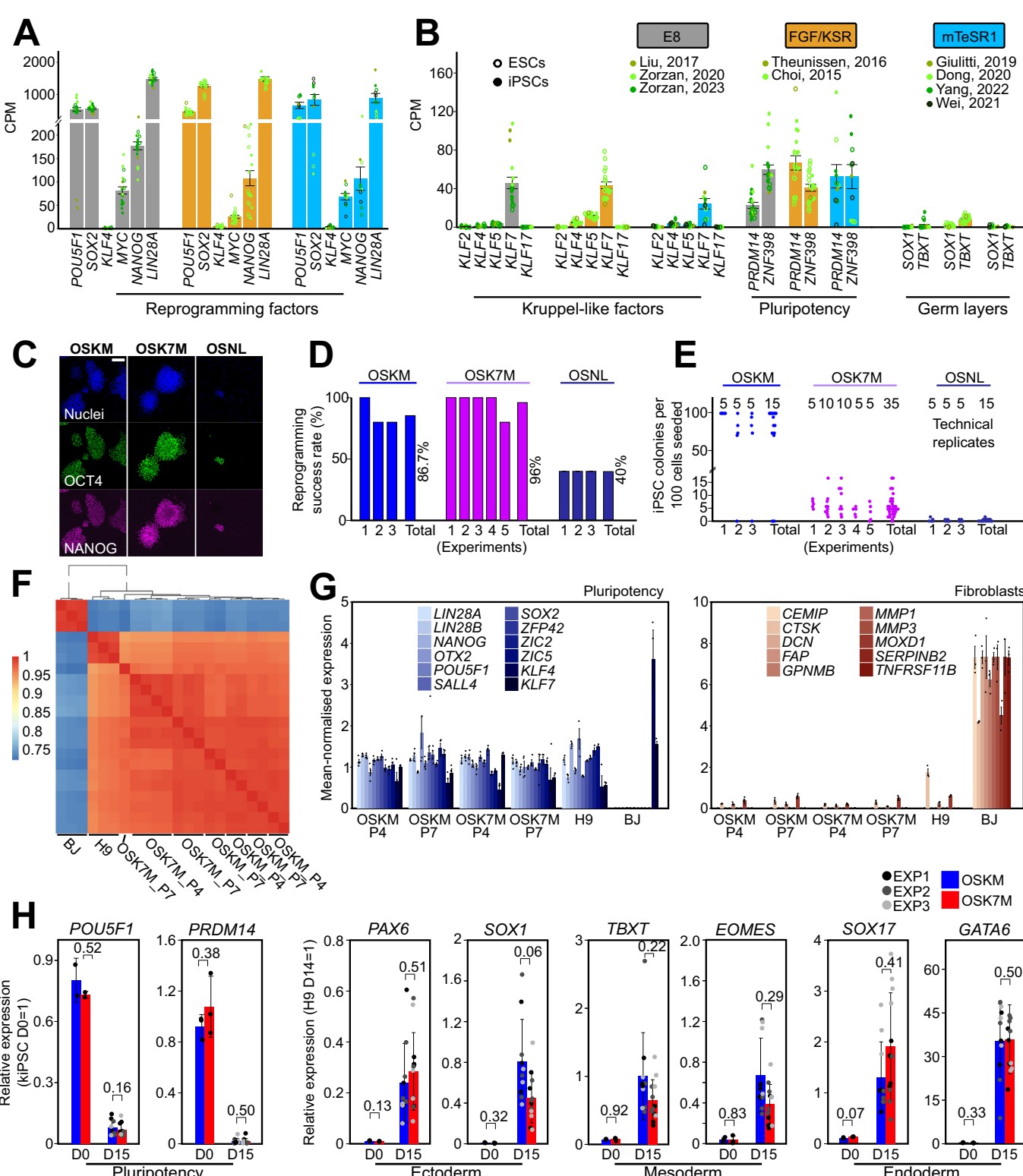

We generated EMPTY and KLF7 iPSCs stably expressing the EOS reporter, in which GFP is expressed under the control of OCT4-SOX2-responsive elements (Hotta et al, 2009), thus serving as a proxy for pluripotency. Overexpression of KLF7 led to a strong

and rapid activation of EOS reporter during chemical resetting, as compared to EMPTY-iPSCs (Fig. 3B).

Transcriptional analysis on the obtained naive iPSCs (niPSCs) showed increased expression of naive and shared pluripotency

◀

**Figure 1. Reprogramming with the OSK7M cocktail generates bona fide human iPSCs.**

(A) Barplot showing the absolute expression, measured by RNAseq, of reprogramming factors in conventional hPSCs cultures in 3 distinct culture conditions. Data were obtained from (Choi et al, 2015; Dong et al, 2020; Giulitti et al, 2019; Jang et al, 2022; Liu et al, 2017; Theunissen et al, 2016; Wei et al, 2021; Zorzan et al, 2020, 2023). Mean +/− SD of at least eight biological replicates. (B) Barplot showing the absolute expression, measured by RNAseq, of Krüppel-like factors, pluripotency and differentiation genes in conventional hPSCs, cultured in three distinct culture conditions. Data were obtained from (Choi et al, 2015; Dong et al, 2020; Giulitti et al, 2019; Jang et al, 2022; Liu et al, 2017; Theunissen et al, 2016; Wei et al, 2021; Zorzan et al, 2020, 2023). Mean +/− SD of at least eight biological replicates. (C) Immunofluorescence images of pluripotency markers OCT4 and NANOG. Nuclei were stained with DAPI. Representative images of at least three independent experiments are shown. Scale bar: 100 μm. (D) Reprogramming success rate calculated as the percentage of technical replicates that produced at least one fully reprogrammed colony. At least three independent experiments are shown. (E) Number of iPSCs colonies obtained at day 14 from 100 cells seeded, by using three distinct reprogramming cocktails. Data from at least three independent experiments, each one conducted in five or ten technical replicates (dots), as indicated in the figure. (F) Heatmap of unsupervised clustering based on transcriptome analysis in primed PSCs (H9), fibroblasts (BJ) and iPSCs obtained from reprogramming of fibroblasts with OSKM and OSK7M and stabilised for four and seven passages. For each condition, two clones at two different passages were analysed. The data points represent technical replicates from a single clone. The colours indicate the correlation index (r). (G) Barplots showing the mean-normalised expression of pluripotency and fibroblast markers in iPSCs stabilised for four and seven passages after reprogramming with OSKM or OSK7M. H9 PSCs and fibroblasts (BJ) were used as controls. Mean +/− SD of at least three biological replicates, shown as dots. (H) Barplots showing relative mRNA expression measured by qPCR of pluripotency and lineage markers in Embryoid bodies (EBs) obtained from differentiation of iPSCs generated by reprogramming with OSKM and OSK7M cocktails and stabilised in culture for 15 passages. Expression was normalised to undifferentiated kiPSCs or to EBs obtained from H9. Means and SD of at least three biological replicates, shown as dots, from three independent experiments, shown in different shades of grey. Unpaired two-tailed t test. Source data are available online for this figure.

markers in KLF7-niPSCs compared to EMPTY-niPSCs, at levels comparable to established niPSCs HPD06 (Giulitti et al, 2019), while the primed marker ZIC2 was not expressed in both lines (Fig. 3C).

We recently reported that chemical resetting generates a mixed population of naive pluripotent cells and trophoblast cells (Zorzan et al, 2023). Whole transcriptome analysis revealed that EMPTY-niPSCs expressed high levels of trophoblast markers, retained low expression of conventional pluripotency markers (e.g. SOX11 and ZIC2) and failed to fully activate naive genes. In stark contrast, KLF7-niPSCs showed full activation of a naive transcriptional programme with barely detectable expression of conventional pluripotency and trophoblast markers (Fig. 3D).

A global overview by Principal Component Analysis (PCA) revealed a similar global expression profile of KLF7-niPSCs compared to two naive iPSC lines (HPD01 and HPD06 (Giulitti et al, 2019)) and distinct from conventional PSCs (H9, HPD00, EMPTY-iPSCs, KLF7-iPSCs) (Fig. 3E). Finally, after prolonged culture, KLF7-niPSCs expanded robustly in naive medium for ten passages while retaining elevated expression of naive markers (Fig. EV3A,B). Moreover, they also displayed comparable developmental potential, measured by the capacity to differentiate to trophoblast stem cells (TSCs) (Fig. EV3A,B).

Taken together, our findings further endorse the observation that expression of KLF7 increases the efficiency of chemical resetting, promoting expression of naive pluripotency genes.

KLF7 and KLF4 showed comparable capacity of somatic reprogramming, so we tested the efficiency of both factors in the context of resetting. To this aim, we first generated conventional PSCs expressing both KLF factors at similar levels (Fig. EV3C). Overexpression of KLF7 gave rise to a morphologically homogeneous population of naive colonies as expected (black arrows), while chemical resetting of cells overexpressing KLF4 generated a more heterogeneous cell population (Fig. 3F). Gene expression analysis showed that conventional markers (OTX2 and ZIC2) were equally repressed in both lines during resetting (Fig. EV3D). In contrast, both the general pluripotency markers NANOG, POU5F1 and ZNF398, and the naive markers TFCP2L1 and DPPA5 were induced at higher levels at day 14 by KLF7 compared to KLF4. In summary, both KLF4 and KLF7 promote morphological and molecular changes associated with naive pluripotency.

## Loss of KLF7 impairs chemical resetting

KLF7 is transiently upregulated during the first days of chemical resetting and its overexpression maximises the acquisition of naive pluripotency identity. We thus asked whether KLF7 might also be required for resetting.

First, we tested the effects of KLF7 inactivation on the maintenance of conventional PSCs by employing clustered regularly interspaced palindromic repeats interference (CRISPRi) technology. We observed a significant reduction in KLF7 mRNA with two independent gRNAs, to ~40% of the levels of parental PSCs (Fig. 4A). Cells could be expanded readily, without signs of differentiation and no effects on the expression of functional pluripotency markers (Zorzan et al, 2020; Wang et al, 2012) (Fig. 4B). We concluded that KLF7 is not required for the maintenance of conventional PSCs.

We then used the same CRISPRi lines for chemical resetting (Fig. 4C), in which naive-like colonies display a round, dome-shaped morphology (Fig. 4D) and co-expression of NANOG and the naive marker SUSD2 (Bredenkamp et al, 2019) (Fig. 4E,F). In the presence of a Control gRNA, naive-like colonies formed efficiently (Fig. 4E–G) and contained between 69.1 and 83.8% of cells co-expressing NANOG and SUSD2 (Fig. EV4). KLF7 inactivation led to a strong reduction in the number of naive-like colonies at day 7 (Fig. 4E) and 14 (Fig. 4D–G), with no significant differences in the fraction of NANOG/SUSD2-positive cells (Fig. EV4). We conclude that endogenous KLF7 promotes chemical resetting.

## KLF7 promotes the acquisition of naive identity

We then asked how, mechanistically, KLF7 promotes efficient resetting. We reasoned that it could either activate naive genes and/ or block the expression of extraembryonic lineages (i.e., trophoblast). Previous transcriptional bulk analyses do not allow to discriminate whether expression of specific markers coexists in the same cells and a single-cell analysis of naive and trophoblast cells is needed. To do so, we applied the chemical resetting protocol and after 3 days of HDAC inhibition, MEK inhibition and LIF stimulation, we exposed cells to either PXGL medium to induce

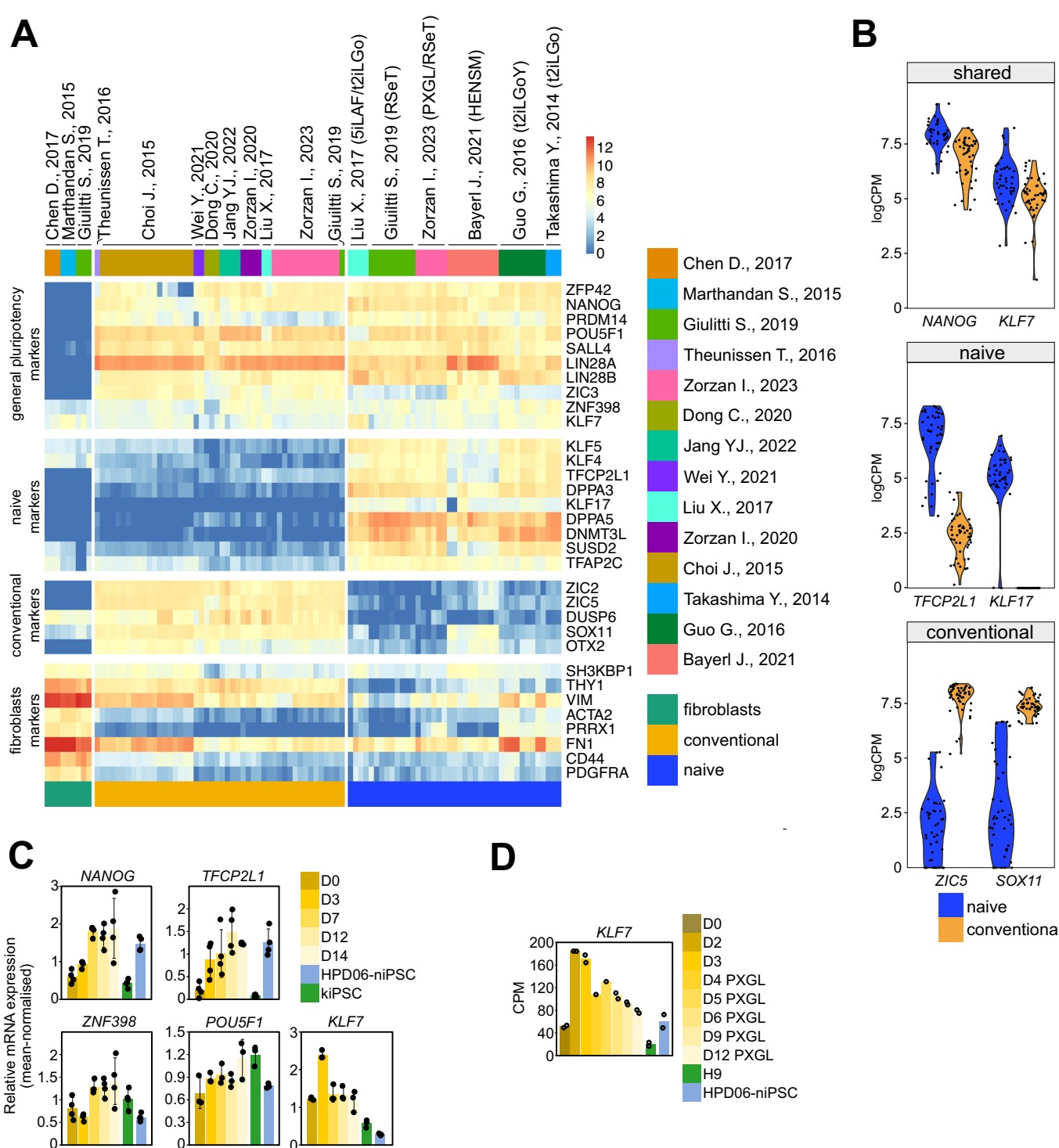

**Figure 2. KLF7 expression in human PSCs.**

(A) Heatmap showing expression, measured by RNAseq (logCPM), of markers for naive hPSCs, conventional hPSCs, general pluripotency and fibroblasts in naive and conventional hPSCs and fibroblast cells. Data were obtained from the indicated publications. (B) Violin plot showing expression, measured by RNAseq, of NANOG, KLF7, TFCP2L1, KLF17, ZIC5 and SOX1 in naive and conventional hPSCs. Data were obtained from publications indicated in (A). (C) Barplots showing expression measured by qPCR of naive and primed pluripotency markers during chemical resetting of kiPSC WT cell line. Naive iPSCs (HPD06) and kiPSC WT cell lines were used as controls. Mean +/− SD of at least three independent experiments, shown as dots. (D) Barplot showing the absolute expression, measured by RNAseq, of KLF7 in kiPSC cells during chemical resetting. H9 hESCs and naive iPSCs (HPD06) were used as controls. Data were obtained from Zorzan et al, 2023. Mean of two biological replicates. Source data are available online for this figure.

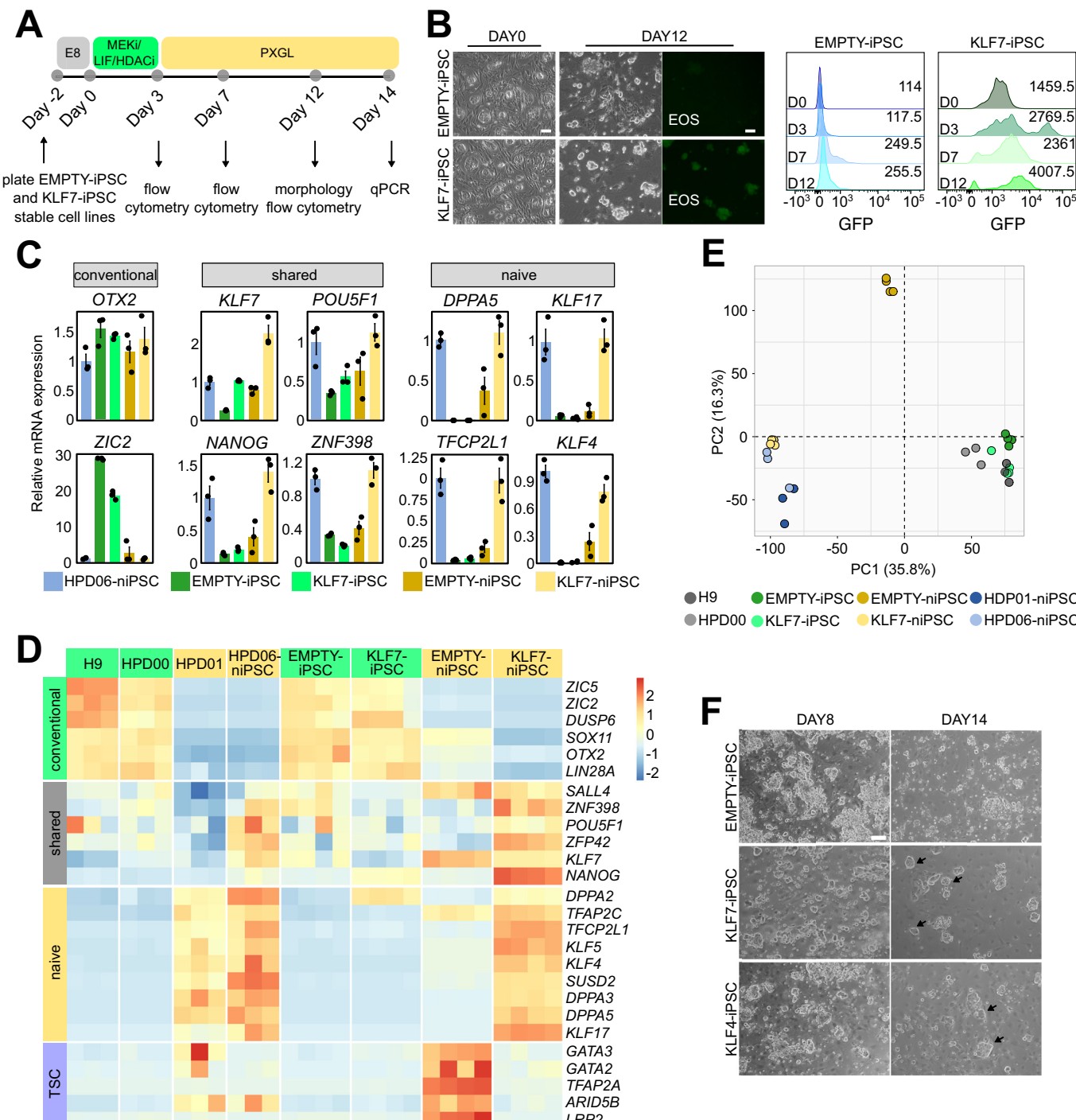

naive pluripotency or TSC medium to drive cells towards the extraembryonic fate (Zorzan et al, 2023) (Fig. 5A,B). We performed quantitative IF staining for the general pluripotency marker OCT4 (Hay et al, 2004; Matin et al, 2004; Wang et al, 2012), the trophoblast marker GATA3 (Dong et al, 2022) and the naive PSCs marker KLF17 (Lea et al, 2021), to monitor cell fate transitions. Conventional PSCs at day 0 (EMPTY-iPSCs day 0) express OCT4 in 100% of cells and progressively acquire naive or trophoblast identity by day 12 (17.5% of KLF17+/GATA3− cells at day 12 in

PXGL medium and 75.5% of GATA3+/KLF17− cells at day 12 in TSC medium, Figs. 5C,D and EV5A). Of note, OCT4 expression is maintained in PXGL medium (80.5% at day 12) and completely abolished in TSC medium (2% at day 12). Interestingly, cells overexpressing KLF7 display more robust activation of the naive marker KLF17 (69.5% of KLF17+/GATA3− cells at day 12 in PXGL medium) and a reduced formation of GATA3+/KLF17− cells in TSC medium (44.5% at day 12). OCT4 expression was maintained in the majority of cells exposed to PXGL (86.5% at day

◄ **Figure 3. KLF7 boosts the efficiency of chemical resetting.**

(A) Schematic representation of the experimental strategy used to chemically reset conventional iPSCs (stably expressing KLF7 or an EMPTY vector) to naive pluripotency. (B) Left: Representative images of conventional hiPSCs overexpressing an empty vector (EMPTY-iPSCs) or the KLF7 transgene (KLF7-iPSCs) at day 0 and after 12 days of chemical resetting. Fluorescence images of OCT4-SOX2-GFP (EOS) reporter are shown. Representative images of three independent experiments are shown. Scale bar: 100 μm. Right: Representative flow-cytometry plots of EOS signal in kiPSC EMPTY and kiPSC KLF7 cell lines at day 0 and after 3, 7 and 12 days of chemical resetting. Representative plots of two independent experiments are shown. Average values of median GFP intensity of two independent experiments are indicated in the right part of each panel. (C) Barplots showing expression measured by qPCR of naive and conventional pluripotency markers in EMPTY-niPSCs and KLF7-niPSCs at day 14, obtained from chemical resetting of kiPSC EMPTY and kiPSC KLF7 cell lines. Naive iPSCs (HPD06) and kiPSC KLF7 and kiPSC EMPTY cell lines were used as controls. Mean +/− SD of three independent experiments, shown as dots. (D) Heatmap of Conventional, Shared and Naive pluripotency genes of samples shown in (E). Z-Scores of row-scaled expression values (CPM) are shown. (E) Principal Component Analysis obtained from analysis of RNAseq data of niPSC EMPTY and niPSC KLF7 cells. At least three biological replicates (dots) were analysed. (F) Bright-field images of conventional hiPSCs overexpressing an empty vector (EMPTY-iPSCs), the KLF7 transgene (KLF7-iPSCs) or the KLF4 transgene (KLF4-iPSCs) after 8 and 12 days of chemical resetting. Representative images of three biological replicates. Scale bar: 200 μm. Source data are available online for this figure.

7 and 89.5% at day 12) and, surprisingly, also in cells exposed to TSC medium (94.5% at day 7 and 12% at day 12).

To further investigate if the mechanism of action of KLF7 in promoting pluripotency genes and blocking GATA3 is direct or indirect, we analysed the transcriptome of conventional PSCs overexpressing KLF7, as several trophoblast markers are already expressed in conventional PSCs (Zorzan et al, 2023) (e.g. GATA2/3, KRT7 and TFAP2A), thus, if KLF7 is a direct repressor of trophoblast markers, we should detect a reduction in their expression in conventional PSCs overexpressing KLF7. However, we observed induction of naive genes (e.g. KLF5, DPPA2, TFAP2C) and of NANOG, repression of the conventional markers ZIC2, but no significant changes in trophoblast markers (Fig. 5E). Of note, TFAP2C has been shown to be required for maintenance and induction of naive pluripotency (Pastor et al, 2018), KLF5 for maintenance of naive pluripotency (Ai et al, 2022), and NANOG for both maintenance and induction of pluripotency (Theunissen et al, 2014).

Altogether, these results indicate that elevated expression of KLF7 promotes the expression of naive pluripotency genes involved in maintenance and induction of pluripotency, but it has no overt effect on trophoblast genes.

In the present study, we investigated the role of KLF7 in the induction of pluripotency. We showed that KLF7 is expressed in both conventional and naive PSCs and when overexpressed in conventional PSCs it enhances chemical resetting to naive PSCs while sustaining OCT4 expression.

Chemical resetting generates an initial plastic state in which naive- and TSC- specific markers are co-expressed, allowing the acquisition of pluripotent or extraembryonic fates depending on the environmental signals to which cells are later exposed (Zorzan et al, 2023). Interestingly, this state is characterised by a peak in KLF7 expression. We showed that KLF7 overexpression during chemical resetting promotes the acquisition of the naive fate, and its sustained expression stabilises naive identity, while its down-regulation results in a severe reduction in resetting efficiency. Finally, KLF7 drives the acquisition of pluripotent identity also during somatic cell reprogramming and in the context of chemical resetting to human naive PSCs.

Our comparison of KLF4 and KLF7 during reprogramming and resetting indicated that utilisation of either transcription factor is a viable approach, but our data suggest that their function may not be exactly identical. In fact, we observed that KLF4 shows higher reprogramming efficiency while KLF7 leads to the formation of

morphologically more homogenous colonies with higher expression of some general and naive pluripotency markers. The reasons for these differences are currently not clear, because KLF4 and KLF7 possess very similar structures and their DNA-binding domains have nearly identical residues (Veerapandian et al, 2018). One possibility to explain this is that KLF4 and KLF7 may interact with distinct sets of proteins. In this regard, available protein-protein interaction databases support this hypothesis (Oughtred et al, 2019), but further studies with KLF4 and KLF7 mutants will be required to test this systematically. It remains to be tested whether utilisation of either KLF4 or KLF7 is preferable in specific assays, for instance, blastoid formation. Human blastoids have been generated with several approaches and utilising human naive ESCs cultured in different conditions, however, in all these protocols a very small number of blastoids develop to a post-implantation-like stage, and even then, some cell types present in human post-implantation embryos are missing (Xie et al, 2025). It would be interesting to test whether our naive ESCs reset with KLF7 could yield blastoids that develop with higher efficiency or can form additional post-implantation cell types.

Krüppel-like factors have been extensively studied in naive mESCs. Klf2, Klf4 and Klf5 have been characterised for their role in reprogramming of somatic, in resetting of primed cells (Dunn et al, 2019; Nakagawa et al, 2008; Jeon et al, 2016) and in maintenance of pluripotency (Yamane et al, 2018). Only the combinatorial depletion of the Klf2/4/5 appears to be detrimental in the maintenance of naive pluripotency, and their overlapping functions target multiple naive-specific transcription factors.

In mouse primed pluripotent cells, Klf2/4/5 are expressed at negligible levels (Fig. EV5B) (Fan et al, 2020; Zhang et al, 2016). Transcriptional analyses of early mouse embryos confirmed expression of Klf2/4/5 only in naive pluripotent cells of the ICM or in naive pre-implantation epiblast (Guo et al, 2010; Boroviak et al, 2015; Ema et al, 2008).

Of note, KLF7 is not expressed at significant levels in murine PSCs (Fig. EV5B), and its forced expression in primed EpiSCs is inconsequential (Zorzan et al, 2020), indicating that KLF7 functions as a pluripotency regulator, is not conserved in rodents. Thus, in murine PSCs, biologically active KLF factors have been found only in the naive state, and future studies will be needed to identify KLFs functionally relevant for murine primed pluripotency.

In human pluripotent stem cells, KLF4, KLF5, KLF7 and KLF17 are expressed in the naive epiblast and in in vitro cultured naive

PSCs (Guo et al, 2016; Giulitti et al, 2019; Liu et al, 2017; Takashima et al, 2014; Blakeley et al, 2015; Zorzan et al, 2023), and act as inducers of naive pluripotency.

Interestingly, KLF7 is also expressed in conventional PSCs, and it promotes both their maintenance (Zorzan et al, 2020) and their generation from somatic cells via reprogramming (Fig. 1).

Thus, in human PSCs, a single KLF acts as a general regulator of pluripotency, similarly to NANOG. NANOG is indeed expressed both in naive and primed pluripotent cells, both in vivo and in vitro, and it promotes both somatic cell reprogramming and resetting. It remains to be seen whether KLF7, similarly to other pluripotency factors, also plays a role in human germ cell specification.

Recently, a stem cell zoo of conventional PSCs of 6 different mammalian species has been generated (Lázaro et al, 2023). It would be interesting to employ those pluripotent cell lines to analyse the evolutionary conservation of KLF factors in the maintenance of pluripotency and in resetting to the naive state.

# Methods

### Reagents and tools table

| Reagent/resource | Reference or source | Identifier or catalogue number |
| --- | --- | --- |
| **Experimental models** | | |
| Human primed iPSCs | Takashima et al, 2014 | kiPSCs |
| Human primed iPSCs – KLF7 | Zorzan et al, 2020 | |
| Human primed iPSCs | Giulitti et al, 2019 | HPD00 |
| Human primed ESCs | WiCell | WA09 |
| Human keratinocytes | Takashima et al, 2014 | |
| Human naive iPSCs | Giulitti et al, 2019 | HPD06 |
| Human naive iPSCs | Giulitti et al, 2019 | HPD01 |
| Human trophoblast stem cells | Zorzan et al, 2023 | |
| **Recombinant DNA** | | |
| pBase | Gift of Austin Smith lab | |
| pENTR2B donor vector | Invitrogen | |
| PB-CAG-DEST-bghpA | Gift of Austin Smith lab | |
| pGK-Hygro | Gift of Austin Smith lab | |
| KLF4-containing plasmid | Addgene | 60435 |
| KLF7-containing plasmid | Zorzan et al, 2020 | |
| CRISPRi *piggyBac* plasmid | Tang et al, 2022 | |

| Reagent/resource | Reference or source | Identifier or catalogue number |
| --- | --- | --- |
| **Antibodies** | | |
| Mouse anti-OCT4 | Santa Cruz, monoclonal | sc-5279 (1:200) |
| Rabbit anti-NANOG | Cell Signaling Technologies, monoclonal | D73G4 (1:100) |
| Goat anti-GATA3 | R&D, polyclonal | AF2605 (1:100) |
| Rabbit anti-KLF17 | Atlas Antibodies, polyclonal | HPA024629 (1:100) |
| Mouse anti-SUSD2 | Miltenyi Biotec | 130127902 (1:200) |
| Donkey anti-Goat IgG (H + L) Secondary Antibody Alexa Fluor 488 | ThermoFisher | A-11055 (1:500) |
| Anti-Mouse IgG (H + L) Secondary Antibody Alexa Fluor 568 | ThermoFisher | A-10037 (1:500) |
| Donkey anti-Rabbit IgG (H + L) Secondary Antibody Alexa Fluor 647 | ThermoFisher | A -31573 (1:500) |
| **Oligonucleotides and other sequence-based reagents** | | |
| NM-microRNAs | Stemgent StemRNA-NM Reprogramming Kit | Reprocell, 00-0076 |
| OCT4 | Individual modified mRNAs | In-house |
| SOX2 | Individual modified mRNAs | In-house |
| NANOG | Individual modified mRNAs | In-house |
| LIN28A | Individual modified mRNAs | In-house |
| KLF7 | Individual modified mRNAs | In-house |
| KLF4 | Individual modified mRNAs | In-house |
| cMYC | Individual modified mRNAs | In-house |
| ACTA2 | Fw: GGAAAAGAT CTGGCACCACT Rev: GAGTCATTT TCTCCCGGTTG | This study |
| CER1 | Fw: GAAATGCGG GTCTGTTCATT Rev: AGTGCATC GTGGTGAACTTG | This study |
| DPPA5 | Fw: AAGTGGAT GCTCCAGTCCAT Rev: ATCCAAGG GCCTAGTTCGAG | This study |
| EOMES | Fw: CCTTCTCAGA AACGCAATTCA Rev: TTGTCTCTG AAGCCTTTTGC | This study |

| Reagent/resource | Reference or source | Identifier or catalogue number |
|---|---|---|
| GAPDH | Fw: CGAGATCCCTCCAAAATCAA<br>Rev: GGCAGAGATGATGACCCTTT | This study |
| GATA6 | Fw: GCAAAAATACTTCCCCCACA<br>Rev: TCTCCCGCACCAGTCATC | This study |
| KLF4 | Fw: CCCAATTACCCATCCTTCCT<br>Rev: CAGGTGTGCCTTGAGATGG | This study |
| KLF7 | Fw: CTCATGGGAGGGATGTGAGT<br>Rev: ACCTGGAAAAACACCTGTCG | This study |
| KLF17 | Fw: CACACAGGTGAGAGGCCATA<br>Rev: TATGCGGGTACACACCAGAT | This study |
| MESP1 | Fw: CTGTTGGAGACCTGGATGC<br>Rev: CGTCAGTTGTCCCTTGTCAC | This study |
| NANOG | Fw: TTTGTGGGCCTGAAGAAAACT<br>Rev: AGGGCTGTCCTGAATAAGCAG | This study |
| NES | Fw: AGGAGAAACAGGGCCTACAGA<br>Rev: GGAGGGTCCTGTACGTGGC | This study |
| OTX2 | Fw: CAAAGTGAGACCTGCCAAAAGA<br>Rev: TGGACAAGGGATCTGACAGTG | This study |
| PAX3 | Fw: TCCACAAGCTGTGTCAGATCC<br>Rev: GCGTTGGAAGGAATCGTGCT | This study |
| PAX6 | Fw: TGGGCAGGTATTACGAGACTG<br>Rev: ACTCCCGCTTATACTGGGCTA | This study |
| POU5F1 / OCT4 | Fw: GTGGAGGAAGCTGACAACAA<br>Rev: ATTCTCCAGGTTGCCTCTCA | This study |
| PRDM14 | Fw: GAGCCTTCAGGTCACAGAGC<br>Rev: TCCACACAGGGGGTGTACTT | This study |
| SOX1 | Fw: GCTGACACCAGACTTGGGTTT<br>Rev: CCCCTCGAGCAAAGAAAACG | This study |
| SOX2 | Fw: GCCGAGTGGAAACTTTTGTCG<br>Rev: GGCAGCGTGTACTTATCCTTCT | This study |
| SOX17 | Fw: ACGCCGAGTTGAGCAAGA<br>Rev: TCTGCCTCCTCCACGAAG | This study |
| TBXT / T | Fw: TATGAGCCTCGAATCCACATAGT<br>Rev: CCTCGTTCTGATAAGCAGTCAC | This study |
| TFCP2L1 | Fw: GGAGTTCCAGCCATGCTCTT<br>Rev: CCTGCTTGAAGATGGGCAGA | This study |
| ZIC2 | Fw: CATGCACGGTCCACACCTC<br>Rev: CTCATGGACCTTCATGTGCTT | This study |
| ZNF398 | Fw: TGGCAAGAATCTCAGCCAAGA<br>Rev: GTGGAGTAAAGTGCTTAGGGC | This study |
| dN6 primers | DN6_A: NNNNNA<br>DN6_T: NNNNNT<br>DN6_G: NNNNNG<br>DN6_C: NNNNNC | This study |
| Control gRNA | Fw: GGAGACGATGCGTAGGGAGATCCGGAATCTATTGGCCCGTCTCC<br>Rev: GGAGACGGGCCAATAGATTCCGGATCTCCCTACGCATCGTCTCC | This study |
| gRNA KLF7_1 | Fw: TGCGCGGAGAACCGAACGGA<br>Rev: TCCGTTCGGTTCTCCGCGCA | This study |
| gRNA KLF7_2 | Fw: GAAGGGAATTGTTACTCCCG<br>Rev: CGGGAGTAACAATTCCCTTC | This study |
| Chemicals, enzymes and other reagents | | |
| Growth factor-reduced Matrigel | Corning | 356231 |
| PBS with MgCl₂/CaCl₂ | Sigma-Aldrich | D8662 |
| E8 medium | Chen et al, 2011 | Made in-house |
| E6 medium | Chen et al, 2011 | Made in-house |
| EDTA | Invitrogen | AM99260G |
| PBS without MgCl₂/CaCl₂ | Sigma-Aldrich | D8537 |
| Mitotically inactivated mouse embryonic fibroblasts | ATCC | DR4 |
| PXGL medium | Bredenkamp et al, 2019 | In-house |
| RSet medium | Stem Cell Technologies | 05978 |

| Reagent/resource | Reference or source | Identifier or catalogue number |
|---|---|---|
| DMEM/F12 | Gibco | 11320074 |
| Neurobasal | Gibco | 21103049 |
| N2 Supplement | Gibco | 17502048 |
| B27 Supplement | Gibco | 17504044 |
| L-glutamine | Gibco | 25030024 |
| 2-mercaptoethanol | Sigma-Aldrich | M3148 |
| PD0325901 | Axon Medchem | Axon 1408 |
| XAV939 | Axon Medchem | Axon 1527 |
| Gö6983 | Axon Medchem | Axon 2466 |
| human LIF | Qkine | Qk036 |
| TrypLE | Gibco | 12563029 |
| ROCK inhibitor Y27632 | Axon Medchem | 1683 |
| Mycoalert | Lonza | |
| Gelatin | Sigma-Aldrich | G1890 |
| Tissue culture plates | Corning | 3516 (6WP); 3513 (12WP) |
| Foetal Bovine Serum | Gibco | A5256701 |
| Penicillin–streptomycin | Gibco | 15140122 |
| Bovine Serum Albumin | Gibco | 15260037 |
| ITS-X | Gibco | 51500056 |
| L-ascorbic acid | Sigma-Aldrich | A4544 |
| EGF | PeproTech | AF-100-15 |
| CHIR99021 | Axon Medchem | 1386 |
| A83-01 | Axon Medchem | 1421 |
| SB431542 | Axon Medchem | 1661 |
| VPA HDACi | Sigma-Aldrich | P4543 |
| Hygromycin B | Thermo Fisher | 10687010 |
| Gateway Cloning Kit | Invitrogen | 11791020 and 11789020 |
| FuGENE HD Transfection | Promega | E2311 |
| Hygromycin B | Invitrogen | 10687010 |
| Vitronectin | Thermo Fisher | A14700 |
| FGF-2 | QKINE | Qk002 |
| KSR | Gibco | 10828028 |
| LSD1i | RN- 1, EMD Millipore | 489479 |
| B18R | Invitrogen | 34-8185-81 |
| StemMACSTM mRNA Transfection Kit | Miltenyi Biotec | 130-104-463 |
| HiScribeTM T7 ARCA mRNA Kit | NEB | E2060S |
| mTeSR | StemCell Technologies | 05850 |
| Ultra low attachment plates | Corning | 3473 |
| Non-essential Amino Acids | Gibco | 11140050 |
| Doxycycline | Sigma-Aldrich | D5207 |
| Trimethoprim | Sigma-Aldrich | T7883 |
| Puromycin | Gibco | A1113803 |

| Reagent/resource | Reference or source | Identifier or catalogue number |
|---|---|---|
| Geneticin | Gibco | 10131027 |
| 4% Formaldehyde | Sigma-Aldrich | 78775 |
| Triton X-100 | Sigma-Aldrich | X100 |
| Horse serum | Gibco | 16050122 |
| DAPI | Sigma-Aldrich | F6057 |
| Hoechst 33342 | ThermoFisher | 62249 |
| Total RNA Purification Kit | Norgen Biotek | 37500 |
| M-MLV Reverse Transcriptase | Invitrogen | 28025-013 |
| SYBR Green Master mix | Bioline | BIO-94020 |
| Quant Seq 3' mRNAseq Library Prep kit | Lexogen | 0152×96 |
| **Software** | | |
| ZEN 2012 | Zeiss | |
| Leica LAS AF (v2.7.3.9723) | Leica | |
| Cell Profiler (v4.1.3) | https://cellprofiler.org/ (Stirling et al, 2021) | |
| QuantStudio™ 6&7 Flex Software 1.0 | Applied Biosystems | |
| BBDuk (BBMap v37.87) | https://archive.jgi.doe.gov/data-and-tools/software-tools/bbtools/bb-tools-user-guide/bbduk-guide/ (Bushnell, 2014) | |
| STAR (v2.7.6a) | https://github.com/alexdobin/STAR (Dobin et al, 2013) | |
| featureCounts (v2.0.1) | https://rdrr.io/bioc/Rsubread/man/featureCounts.html (Liao et al, 2014) | |
| R v4.2.2 | https://www.r-project.org/ (R Core Team, 2022) | |
| Bioconductor (v3.7) | https://bioconductor.org/news/bioc_3_7_release (Huber et al, 2015) | |
| DESeq2 (v1.28.1) | https://bioconductor.org/packages/release/bioc/html/DESeq2.html (Love et al, 2014) | |
| Pheatmap (v1.0.12) | https://cran.r-project.org/web/packages/pheatmap/index.html (Kolde, 2025) | |
| Ggpubr (v0.4.0) | https://rpkgs.datanovia.com/ggpubr/reference/ggpubr-package.html (Kassambara, 2025) | |
| Salmon (v1.6.0 and v1.9.0) | https://salmon.readthedocs.io/en/latest/salmon.html (Patro et al, 2017) | |
| Tximport (v1.20.0 and v1.26.1) | https://bioconductor.org/packages/release/bioc/html/tximport.html (Soneson et al, 2016) | |
| Sva (v3.46.0) | https://code.bioconductor.org/browse/sva/ (Leek and Storey, 2007) | |

| Reagent/resource | Reference or source | Identifier or catalogue number |
|---|---|---|
| edgeR (v3.40.2) | https://bioconductor.org/packages/release/bioc/html/edgeR.html (Chen et al, 2025) | |
| ggplot2 v3.4.2 | https://ggplot2.tidyverse.org/ Wickham H, 2016 | |
| BD FACSDivaTM (v. 9.0) | BDBiosciences | |
| FlowJo software (v10.9.0) | https://www.flowjo.com/ | |
| Fiji 1.0 (ImageJ2) | https://imagej.net/software/fiji/ (Schindelin et al, 2012) | |
| Other | | |
| Zeiss LSM700 | Zeiss | |
| Leica SP8 | Leica | |
| Leica SP5 | Leica | |
| Fluorometer (Qubit) | Qubit | |
| Bioanalyzer (Agilent) | Agilent | |
| NextSeq500 ILLUMINA | ILLUMINA | |
| BD FACSCantoTM II cytometer | BDBiosciences | |

## Culture of hPSCs

Human primed hiPSCs (kiPSCs (Takashima et al, 2014)), Keratinocytes induced Pluripotent Stem Cells, expressing either an empty vector or KLF7 (Zorzan et al, 2020)) and HPD00 (Giulitti et al, 2019) and hESCs (H9) were cultured Feeder-free on pre-coated plates with 0.5% growth factor-reduced Matrigel (CORNING 356231) (vol/vol in PBS with $MgCl_2/CaCl_2$, Sigma-Aldrich D8662) in E8 medium (made in-house according to Chen et al, 2011) at 37 °C, 5% $CO_2$, 5% $O_2$. Cells were passaged every 3–4 days at a split ratio of 1:8 following dissociation with 0.5 mM EDTA (Invitrogen AM99260G) in PBS without $MgCl_2/CaCl_2$ (Sigma-Aldrich D8537), pH8. kiPSCs line was derived by reprogramming human keratinocytes (Invitrogen) with Sendai viruses encoding for OSKM and kindly provided by Austin Smith's laboratory.

Human naive iPSCs (HPD06 and HPD01), previously generated by direct reprogramming of somatic cells and described in (Giulitti et al, 2019) were cultured on mitotically inactivated mouse embryonic fibroblasts (MEFs; DR4 ATCC) in PXGL medium (Bredenkamp et al, 2019) or in RSeT medium (Stem Cell Technologies 05978), at 37 °C, 5% $CO_2$, 5% $O_2$. PXGL medium was prepared as follows: N2B27 (DMEM/F12 [(Gibco 11320074], and Neurobasal in 1:1 ratio [Gibco 21103049], with 1:200 N2 Supplement [Gibco 17502048], and 1:100 B27 Supplement [Gibco 17504044], 2 mM L-glutamine [Gibco 25030024], 0.1 mM 2-mercaptoethanol [Sigma-Aldrich M3148]) supplemented with 1 μM PD0325901 (Axon Medchem), 2 μM XAV939 (Axon Medchem), 2 μM Gö6983 (Axon Medchem) and 10 ng/ml human LIF (Qkine). Human naive PSCs were passaged as single cells every 4 days at split ratio 1:3 or 1:4 following dissociation with TrypLE (Gibco 12563029) for 10 min at room temperature (RT). ROCK inhibitor (Y27632, Axon Medchem 1683) was added in the naive

medium only for 24 h after passaging. All cell lines were mycoplasma negative (Mycoalert, Lonza).

hTSCs were cultured as previously described (Okae et al, 2018). Briefly, cells were cultured in mitotically inactivated mouse embryonic fibroblasts, previously seeded on plastic plates coated with 1% gelatin, at 37 °C in 5% $CO_2$ and 5% $O_2$. TSC medium was prepared as follows: DMEM/F12 supplemented with 0.1 mM 2-mercaptoethanol, 0.2% FBS, 0.5% penicillin–streptomycin, 0.3% BSA [Gibco 15260037], 1% ITS-X [Gibco, 51500056], 1.5 μg/ml L-ascorbic acid [Sigma-Aldrich A4544], 50 ng/ml EGF [PeproTech AF-100-15], 2 μM CHIR99021 [Axon Medchem 1386], 0.5 μM A83-01 [Axon Medchem 1421], 1 μM SB431542 [Axon Medchem 1661], 0.8 mM VPA [HDACi, Sigma-Aldrich, P4543], and 5 μM Y-27632. Media was changed every 2 days and cells were passaged using TrypLE Express every 3–4 days at a ratio of 1:8.

## Generation of human PSCs stably expressing genes of interest

Conventional PSCs stably expressing KLF7, KLF4 or an empty vector were generated via transfection of piggyBAC vectors followed by selection with Hygromycin B (200 μg/ml; Invitrogen 10687010), as described previously (Zorzan et al, 2020).

Briefly, stable transgenic hPSCs expressing KLF7 were generated by transfecting cells with PB transposon plasmids with PB transposase expression vector pBase. In order to generate the PB plasmids, the candidate KLF7 was amplified from cDNA and cloned into a pENTR2B donor vector. Then, the transgene was Gateway cloned into the same destination vector containing PB-CAG-DEST-bghpA and pGK-Hygro selection cassette. KLF4 plasmids obtained from Addgene (#60435), KLF7 generated previously (Zorzan et al, 2020). For DNA transfection, 250,000 hPSCs were dissociated as single cells with TrypLE (Gibco 12563029) and were co-transfected with PB constructs (550 ng) and pBase plasmid (550 ng) using FuGENE HD Transfection (Promega E2311), following the protocol for reverse transfection. For one well of a 12-well plate, we used 3.9 μl of transfection reagent, 1 μg of plasmid DNA, and 250,000 cells in 1 ml of E8 medium with 10 μM Y27632 (ROCKi, Rho-associated kinase (ROCK) inhibitor, Axon Medchem 1683). The medium was changed after overnight incubation, and Hygromycin B (200 μg/ml; Invitrogen 10687010) was added after 48 h. For the overexpression experiments, hPSCs stably expressing an empty vector or the candidates were plated.

## Reprogramming

All reprogramming experiments were performed in microfluidics as previously described (Zorzan et al, 2022) in hypoxia conditions (37 °C, 5% $CO_2$, 5% $O_2$). Microfluidic channels were coated with 25 μg ml−1 Vitronectin (Thermo Fisher, A14700) for 1 h at room temperature (RT) and fibroblasts were seeded at day 0 at 15 cells per mm² in DMEM/10% FBS. On day 1, 9 h before the first mRNAs transfection, E6 medium made in-house according to Chen et al, 2011, including FGF-2 100 ng ml−1 (QKINE Cat. no Qk002 recombinant zebrafish FGF-2), 1% KSR (Gibco, 10828028), ROCKi 5 μM, LSD1i 0.1 μM (RN-1, EMD Millipore Cat. no 489479) and 200 ng ml−1 B18R (Invitrogen 34-8185-81) was applied. The B18R protein was added to the medium to reduce the interferon response.

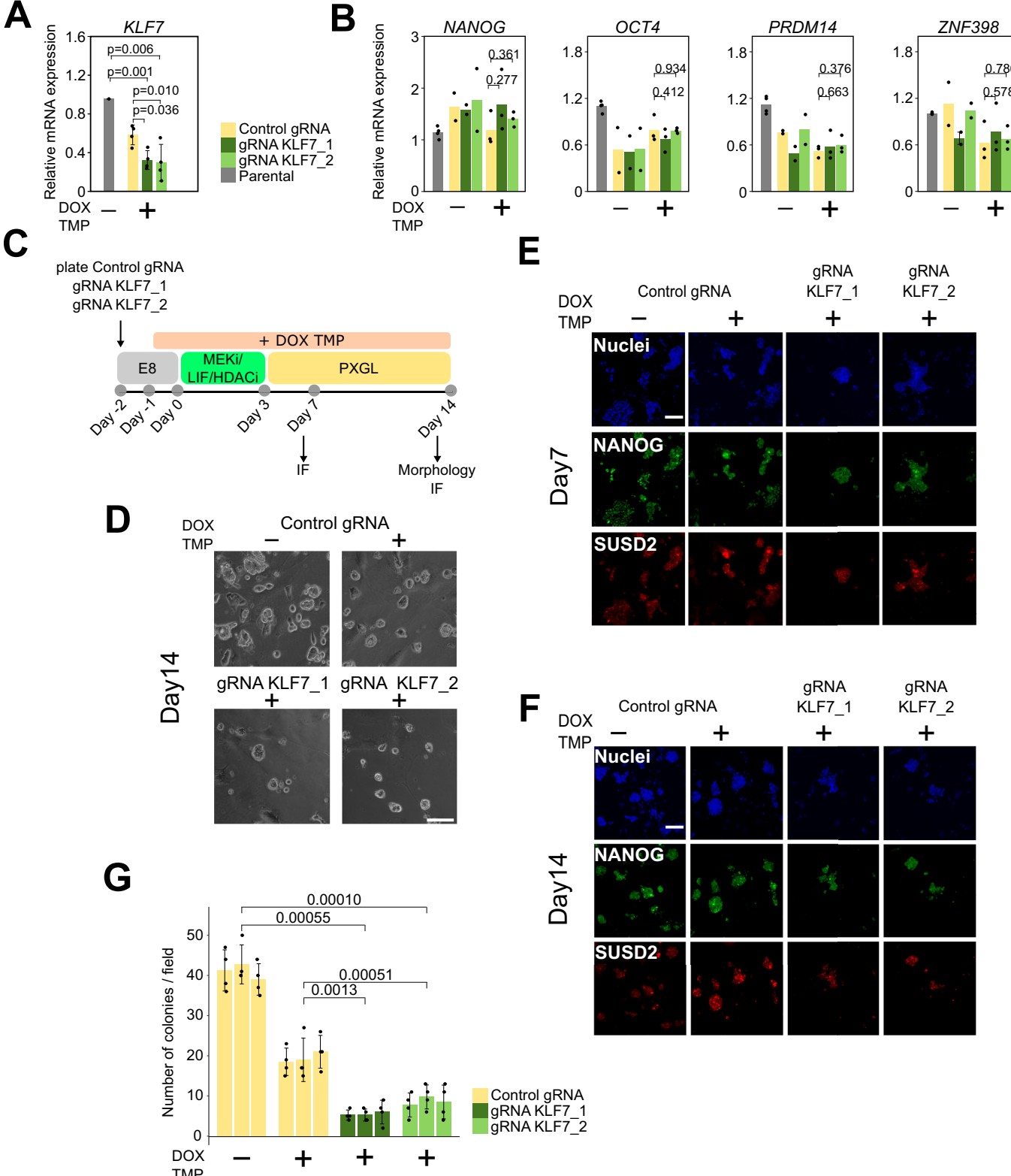

Figure 4. KLF7 promotes the acquisition of naive identity.

(A) Barplots showing expression measured by qPCR of KLF7 in kiPSC CRISPRi lines. Parental kiPSC cells were first transfected with a piggyBAC CRISPRi plasmid (see "Methods") and with a second plasmid encoding a non-targeting control gRNA, or two gRNAs for the KLF7 promoter. Mean +/− SD of 4 biological replicates, shown as dots. "−" indicates vehicle-treated cells, "+" indicates cells treated with DOX and trimethoprim (TMP). Unpaired two-tailed $t$ test. (B) Barplots showing expression measured by qPCR of pluripotency markers in kiPSC CRISPRi lines and parental kiPSCs. Mean of at least two biological replicates, shown as dots. (C) Experimental strategy used to test the role of KLF7 during chemical resetting. (D) Bright-field images at day 14 of chemical resetting. Scale bar: 100 µm. (E, F) Immunofluorescence staining for the general pluripotency marker NANOG and the naive marker SUSD2 at days 7 and 14 of chemical resetting. Scale bar: 100 µm. (G) Quantification based on the morphology of the number of naive colonies observed at day 14 of chemical resetting. $N = 3$ biological replicates (bars), with four technical replicates (dots) each. Unpaired two-tailed $t$ test. Source data are available online for this figure.

Cells were transfected daily at 6 PM and fresh medium was given daily at 9 AM. The transfection mix was prepared according to the StemMACS™ mRNA Transfection Kit (Miltenyi Biotec, 130-104-463) and using modified mRNAs (coding cocktail components) and NM-microRNAs (Stemgent StemRNA-NM Reprogramming Kit). Individual modified mRNAs (OCT4, SOX2, NANOG, LIN28A, KLF7, KLF4, cMYC) were made in-house by in vitro transcription using mRNA synthesis with HiScribe™ T7 ARCA mRNA Kit (NEB E2060S) according to the manufacturer's instructions. During the 8 days of transfection, the dose of mRNAs was gradually increased according to the cell proliferation rate and transfection-induced cell mortality. From day 9, cells were cultured in mTeSR (StemCell Technologies 05850) for 6 days to allow stabilisation of hiPSC colonies. Fresh medium was given twice a day at 9 AM and 6 PM.

## Embryoid bodies differentiation

For Embryoid bodies (EBs) differentiation assay, cells were detached as clumps with EDTA and plated on ultra-low attachment surface plates (CORNING 3473) in E8 medium with 10 µM ROCKi. After 2 days, E8 medium was substituted with DMEM, 20% FBS, 2 mM L-glutamine, 1% NEAA and 0.1 mM 2-mercaptoethanol. Medium was changed every 2 days.

## Chemical resetting

For the chemical resetting from conventional to naive PSC or TSC, kiPSC were seeded at 10,000 cells/cm$^2$ on mitotically inactivated MEFs in E8 medium with 10 µM ROCKi (added only for 24 h). In the case of CRISPRi experiments, 16,000 cells/cm$^2$ were used. Two days after plating (day 0), medium was changed to PD03/LIF/HDACi [N2B27 with 1 µM PD0325901, 10 ng/ml human LIF and 1 mM VPA (HDACi)]. Following 3 days in PD03/LIF/HDACi, medium was changed to PXGL or TSC medium. Cells were passaged with TrypLE Express when confluent (around day 8–9) on MEFs.

## CRISPR interference

For CRISPRi, we used a piggyBac plasmid generated in the Surani lab (Tang et al, 2022) containing a KRAB–dCas9–ecDHFR and a IRES–EGFP fragment downstream of the TRE3G promoter. TRE3G is a Dox-inducible promoter, while DHFR is protein destabilisation degron. The addition of Dox (0.1 µg/ml) and trimethoprim (TMP 10 µM) allows robust mRNA and protein expression of KRAB–dCas9 CRISPRi machinery. Parental conventional PSCs were transfected with the piggyBAC

KRAB–dCas9–ecDHFR plasmid. After selection with Puromycin (0.3 µg/ml; Gibco A1113803) for 5 days, a second plasmid encoding for gRNA was transfected, followed by selection with Geneticin (200 µg/ml; Gibco 10131027). A non-targeting gRNA was used for control, while 2 gRNA recognising the KLF7 promoter were used (see Reagents and Tools Table for sequences). For resetting experiments, we lowered the concentration of Dox (0.025 µg/ml) and TMP (5 µM) to reduce cellular stress.

## Immunofluorescence

Immunofluorescence was performed on 1% Matrigel-coated glass coverslip in wells or in situ in microfluidic channels with the same protocol. For chemical resetting experiments, cells were seeded on MEFs plated on 0.5% Matrigel-coated glass coverslips at least 1 day before. Cells were fixed in 4% Formaldehyde (Sigma-Aldrich 78775) in PBS for 10 min at RT, washed in PBS, permeabilized for 1 h in PBS + 0.3% Triton X-100 (PBST) at RT, and blocked in PBST + 5% of Horse serum (Gibco 16050122) for 5 h at RT. Cells were incubated overnight at 4 °C with primary antibodies (See Reagents and Tools Table) in PBST + 3% of Horse serum. After washing with PBS, cells were incubated with secondary antibodies (Alexa, Life Technologies) (See Reagents and Tools Table) for 45 min at RT in the case of Matrigel-coated glass coverslip and for 2 h for staining in microfluidic devices. Nuclei were stained with either DAPI (4′,6-diamidino-2-phenylindole, Sigma-Aldrich F6057) for glass coverslip or Hoechst 33342 (ThermoFisher 62249) for microfluidic channels. Images were acquired with a Zeiss LSM700, Leica SP8 or Leica SP5 confocal microscopes using ZEN 2012 or Leica TCS SP5 LAS AF (v2.7.3.9723) software, respectively.

For chemical resetting, fluorescence intensity was quantified with Cell Profiler Software (v4.1.3). DAPI staining was used to identify individual cell nuclei and the cytoplasm surrounding each nucleus. Between four and seven independent fields for each sample were quantified. Integrated intensity values were plotted as log (Integrated Intensity+1). Protein presence in the cell population was determined by setting the integrated intensity threshold to 95% of the cumulative density function in samples where the protein is expressed (GATA3 in TSCs, KLF17 and SUSD2 in niPSCs and OCT4 in niPSC and kiPSC). Thresholds used to determine the percentage of positive and negative cells for a given protein were as follows: thr$^{GATA3}$ = 1.895, thr$^{KLF17}$ = 1.8, thr$^{OCT4}$ = 1.74.

## Quantitative PCR

Total RNA was isolated using Total RNA Purification Kit (Norgen Biotek 37500), and complementary DNA (cDNA) was made from

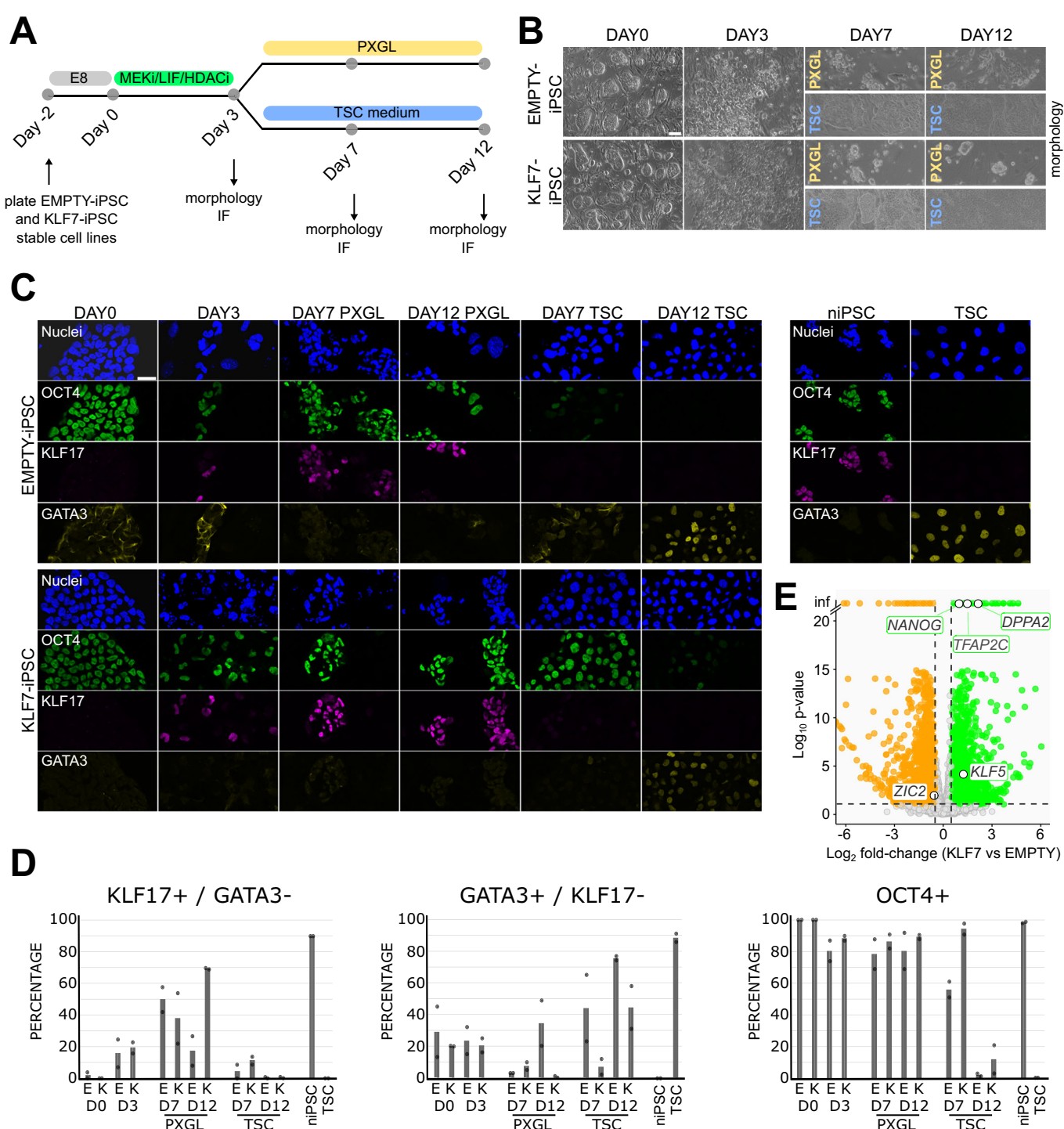

RNA sequencing and analyses

500 ng using M-MLV Reverse Transcriptase (Invitrogen 28025-013) and dN6 primers. For real-time PCR, SYBR Green Master mix (Bioline BIO-94020) was used. Primers are detailed in the Reagents and Tools Table. Three technical replicates were performed for all quantitative PCR experiments. GAPDH was used as an endogenous control to normalise expression. qPCR data were acquired with QuantStudio™ 6&7 Flex Software 1.0.

**RNA sequencing and analyses**

Quant Seq 3′ mRNAseq Library Prep kit (Lexogen) was used for library construction. Library quantification was performed by a fluorometer (Qubit) and bioanalyzer (Agilent). Sequencing was performed on NextSeq500 ILLUMINA instruments to produce 5 million reads (75 bp SE) for the sample. For the analysis of TSCs markers, the reads were

**Figure 5.  KLF7 promotes the acquisition of naive identity.**

(A) Schematic representation of the experimental strategy used for chemical resetting of kiPSC EMPTY or kiPSC KLF7 cells to naive or Trophoblast Stem cells (TSC), by using PXGL or TSC medium. (B) Bright-field images of conventional kiPSC EMPTY and kiPSC KLF7 cells at day 0 and after 3, 7 and 12 days of chemical resetting in PXGL or TSC medium, as shown in Fig. 4A. Representative images of three independent experiments. Scale bar: 100 μm. (C) Immunofluorescence staining for OCT4, KLF17 and GATA3 in kiPSC EMPTY and kiPSC KLF7 cells at day 0 and after 3, 7 and 12 days of chemical resetting in PXGL or TSC medium, as shown in Fig. 4A. niPSCs and TSCs were used as positive controls. Nuclei were stained with DAPI. Representative images of two independent experiments are shown. Scale bar: 30 μm. (D) Barplot showing the percentages of the indicated populations of cells after quantification of data shown in (C). See Fig. EV5A for details. E = EMPTY-iPSCs, K = KLF7-iPSCs. Bars indicate the mean of two independent experiments, shown as dots. (E) Volcano plot showing transcriptome analysis of kiPSC KLF7 cells compared to kiPSC EMPTY cells, in standard culture conditions (E8). Downregulated (Log2 fold change < −0.5 and P value < 0.05) and upregulated (Log2 fold change >0.5 and P value < 0.05) genes are indicated in green and orange, respectively. Known naive pluripotency markers (green) and conventional hPSCs markers (orange) are highlighted. P values were calculated with Wald Test. N = 4 biological replicates for each cell line. Source data are available online for this figure.

trimmed using BBDuk (BBMap v37.87), with parameters indicated in the Lexogen data analysis protocol. After trimming, reads were aligned to the Homo sapiens genome (GRCm38.p13) using STAR (v2.7.6a). The gene expression levels were quantified using featureCounts (v2.0.1). Genes were sorted, removing those that had a total number of counts below 10 in at least 3 samples. All RNA-seq analyses were carried out in the R environment (v4.0.0) with Bioconductor (v3.7). We computed differential expression analysis using the DESeq2 R package (v1.28.1) (Love et al, 2014). Transcripts with an absolute value of log2[FC] > 1 and an adjusted $p$ value < 0.05 (Benjamini–Hochberg adjustment) were considered significant and defined as differentially expressed for the comparison in the analysis. Heatmaps were made using counts-per-million (CPM) values with the pheatmap function from the pheatmap R package (v1.0.12) on differentially expressed genes or selected markers. Volcano plots were computed with log2[FC] and −log10[adjusted $P$ value] from DESeq2 differential expression analysis output using the ggscatter function from the ggpubr R package (v0.4.0). Barplots were made using CPM values with the ggbarplot function from the ggpubr R package. For the time point analysis during resetting and the following stabilisation in PXGL or TSC medium, transcript quantification was performed from raw reads using Salmon (v1.6.0) (Patro et al, 2017) on transcripts defined in Ensembl 105. Gene expression levels were estimated with tximport R package (v1.20.0) (Soneson et al, 2016) and differential expression analysis was computed using the DESeq2 R package (v1.28.1) (Love et al, 2014). Transcripts with an absolute value of log2[FC] ≥ 3 and an adjusted $P$ value < 0.05 (Benjamini–Hochberg adjustment) were considered significant. Principal component analysis was performed on variance stabilised data (vst function from DESeq2 R package v1.32.052) using the prcomp function on the top 5000 most variable genes. Heatmaps were performed using the pheatmap function (pheatmap R package v1.0.12) on log2 count per million (CPM) data of selected markers. All analyses except salmon were performed in R version 4.1.1.

Analyses of published RNA sequencing data were performed as follows:

Raw-reads were downloaded for samples from the relative databases as described in Dataset EV1. Transcript quantification was performed using Salmon (v1.9.0) (Patro et al, 2017) with transcripts defined in Ensembl 106. Gene expression levels were estimated with tximport R package (v1.26.1) (Soneson et al, 2016). Batch correction was performed using ComBat_seq function from the sva R package (v3.46.0; https://bioconductor.org/packages/release/bioc/html/sva.html). Batches have been defined following library preparation: batch 1 for full-length libraries and batch 2 for Quant Seq 3′ mRNAseq Library Prep kit of the GEO series GSE184562.

CPM on batch corrected counts were computed using the cpm function of edgeR package (v3.40.2; (Robinson et al, 2010)). PCA was performed using the svd r function on log transformed CPM.

All plots except the heatmap have been done using ggplot2 v3.4.2 (https://ggplot2.tidyverse.org/authors.html#citation). Heatmap was done using the pheatmap R package (v1.0.12). All analyses were performed using R v4.2.2.

## Flow cytometry

After dissociation in a single-cell suspension using TrypLE, cells were resuspended in PBS and filtered. Cells were analysed according to EOS-GFP expression with BD FACSCantoTM II cytometer and BD FACSDivaTM (v. 9.0) software. Data analysis was performed with FlowJo software (v10.9.0).

## Statistics and reproducibility

For each dataset, sample size n refers to the number of independent experiments or biological replicates, shown as dots, as stated in the figure legends. R software (v4.0.0) was used for statistical analysis. Error bars indicate standard deviation (SD) as stated in the figure legends. $P$ values were calculated using Wald Test or Unpaired two-tailed $t$ test and are reported in the plots. Data analyses have been performed using identical parameters and software. All qPCR experiments were performed with three technical replicates.

# Data availability

Sequencing data that support the findings of this study have been deposited in the Gene Expression Omnibus (GEO) under accession code GSE242158. Previously published RNAseq data that were re-analysed here are available under accession codes listed in Dataset EV1.

The source data of this paper are collected in the following database record: biostudies:S-SCDT-10_1038-S44319-025-00595-2.

# Peer review information

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

## Acknowledgements

We thank the Martello lab for critical feedback on the manuscript. Moreover, we would like to express our gratitude to the NGS facility at the Department of Biology, University of Padua. GA's laboratory is supported by FEBS Excellence award. GM's laboratory is supported by grants from the Giovanni Armenise–Harvard Foundation, the Fondazione Telethon (GJC21157), Progetti di Rilevante Interesse Nazionale PRIN 2022 (Dissecting genetic, epigenetic and metabolic alterations caused by reprogramming of somatic cells to pluripotency), the Microsoft Research Ltd, the European Research Council Starting Grant (MetEpiStem) and the European Union – Next Generation EU, Mission 4 Component 1 CUP C93C22002780006. EC's group is supported by a Progetti di Rilevante Interesse Nazionale PRIN2022 PNRR project (EMBRYODIET).

## Author contributions

**Mattia Arboit**: Conceptualisation; Formal analysis; Visualisation. **Irene Zorzan**: Conceptualisation; Data curation; Formal analysis; Writing—review and editing. **Eleonora Pensabene**: Data curation; Formal analysis. **Marco Pellegrini**: Data curation; Formal analysis; Investigation. **Federica Bertelli**: Data curation; Formal analysis; Investigation. **Giorgia Panebianco**: Data curation; Formal analysis; Methodology. **Davide Benvegnù**: Data curation; Formal analysis; Visualisation. **Giada Rossignoli**: Data curation; Formal analysis; Methodology. **Paolo Martini**: Data curation; Formal analysis; Investigation; Methodology. **Gianluca Amadei**: Visualisation; Writing—review and editing. **Elena Carbognin**: Conceptualisation; Data curation; Formal analysis; Supervision; Funding acquisition; Investigation; Methodology; Writing—original draft; Project administration. **Graziano Martello**: Conceptualisation; Data curation; Formal analysis; Supervision; Funding acquisition; Investigation; Methodology; Writing—original draft; Project administration.

Source data underlying figure panels in this paper may have individual authorship assigned. Where available, figure panel/source data authorship is listed in the following database record: biostudies:S-SCDT-10_1038-S44319-025-00595-2.

## Disclosure and competing interests statement

The authors declare no competing interests.

# Expanded View Figures

**Figure EV1.  Characterisation of OSK7M iPSCs.**

(A) Barplot showing the absolute expression, measured by RNAseq, of Krüppel-like factors KLF4 and KLF7 in conventional hESCs and hiPSCs. Data were obtained from (Choi et al, 2015; Dong et al, 2020; Giulitti et al, 2019; Jang et al, 2022; Liu et al, 2017; Theunissen et al, 2016; Wei et al, 2021; Zorzan et al, 2020, 2023). Mean $+/-$ SD of at least 8 biological replicates is shown. (B) Brightfield images of iPSCs obtained at day 14 from fibroblasts reprogrammed by using 3 distinct reprogramming cocktails, OSKM, OSK7M and OSNL. Dashed circles indicate fully reprogrammed iPSCs colonies. Representative images of at least 3 independent experiments are shown. Scale bars: 100 μm. (C) Representative brightfield images of stabilised iPSCs cultures (10, 15 and 20 passages), obtained with OSKM and OSK7M reprogramming cocktails. Representative images of at least 3 independent experiments are shown. Scale bar: 100 μm. (D) Immunofluorescence images of pluripotency markers OCT4 and NANOG on stabilised iPSCs cultures (15 and 20 passages). Nuclei were stained with DAPI. Representative images of at least 3 independent experiments are shown. Scale bar: 30 μm. (E) Barplots showing expression measured by qPCR of primed pluripotency markers in stabilised iPSCs cultures, obtained with OSKM and OSK7M reprogramming cocktails. Bars= mean of 2 independent experiments, shown as dots. Source data are available online for this figure.

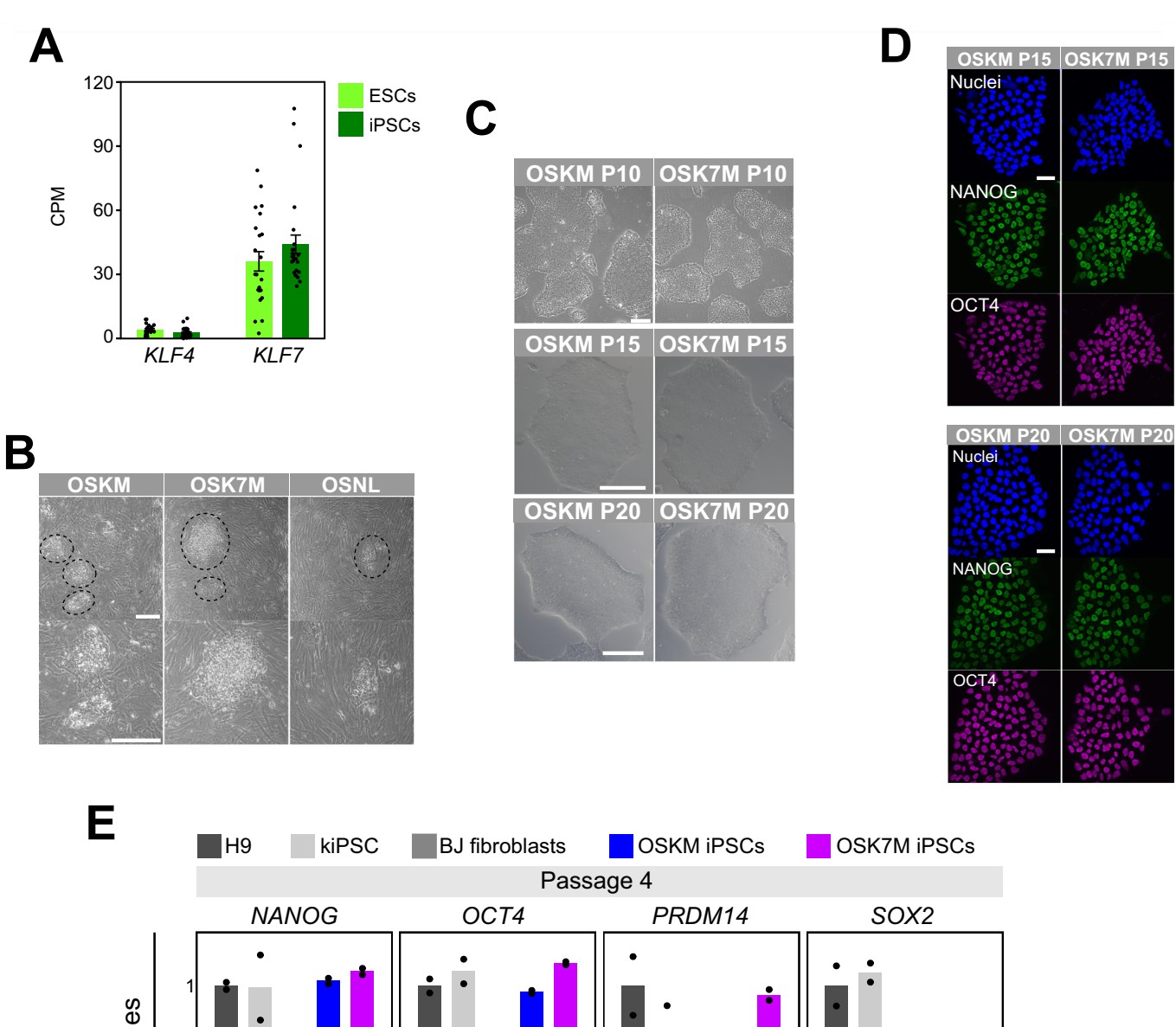

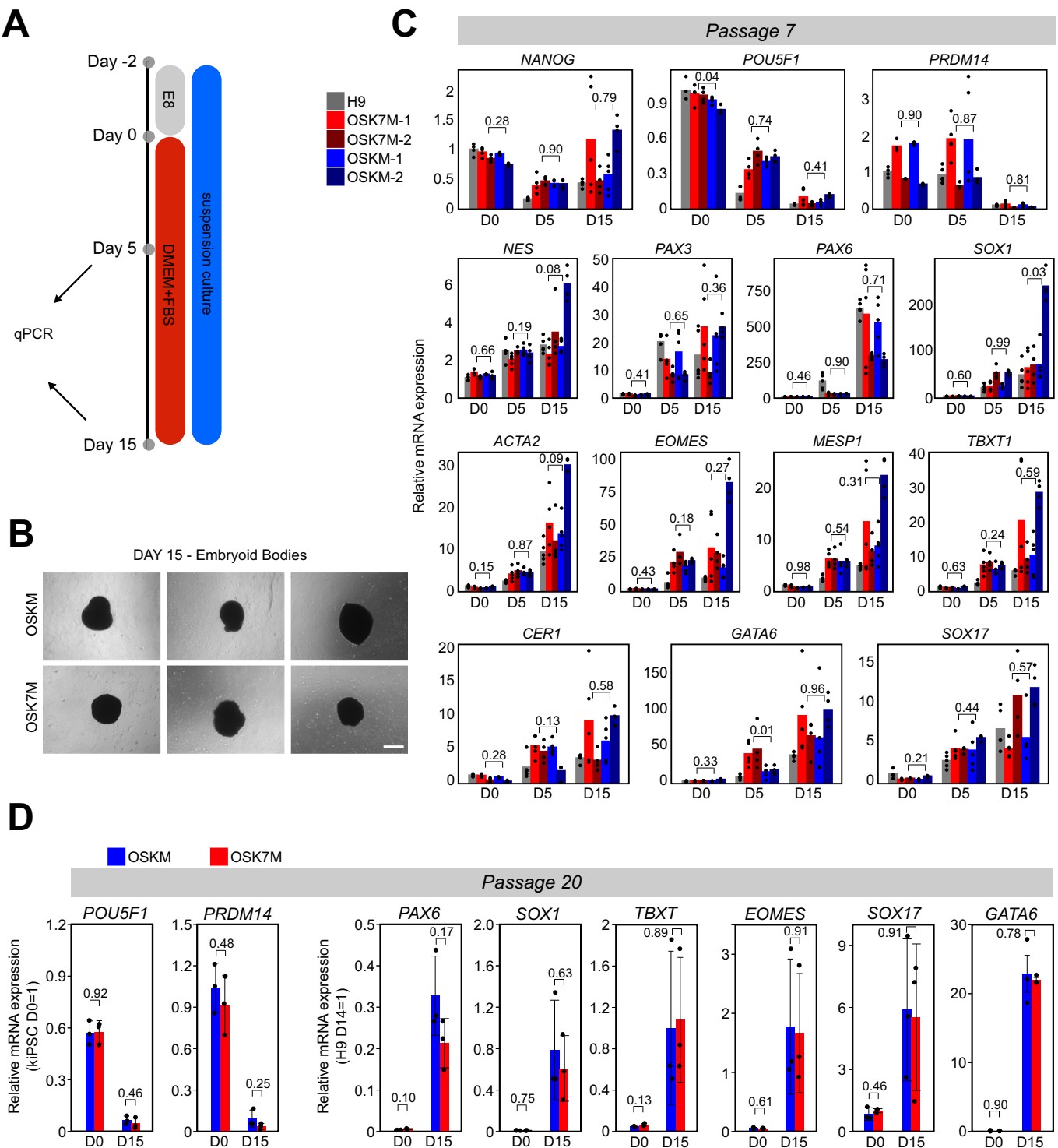

**Figure EV2. Embryoid bodies differention of OSKM and OSK7M iPSCs.**

(A) Schematic representation of experimental strategy used for Embryoid Bodies (EBs) differentiation of iPSCs obtained from reprogramming of fibroblasts with OSKM and OSK7M. (B) Representative bright field images of EBs obtained from OSKM and OSK7M iPSCs. Scale bar = 200 μm. (C, D) Barplots showing relative mRNA expression measured by qPCR of pluripotency and lineage markers in Embryoid bodies (EBs) obtained from differentiation of iPSCs generated by reprogramming with OSKM and OSK7M cocktails and stabilised in culture for 7 and 20 passages. kiPSC and H9 cell lines were used as positive control. Bars indicate the mean of at least 3 biological replicates, shown as dots. Unpaired two-tailed t test. Source data are available online for this figure.

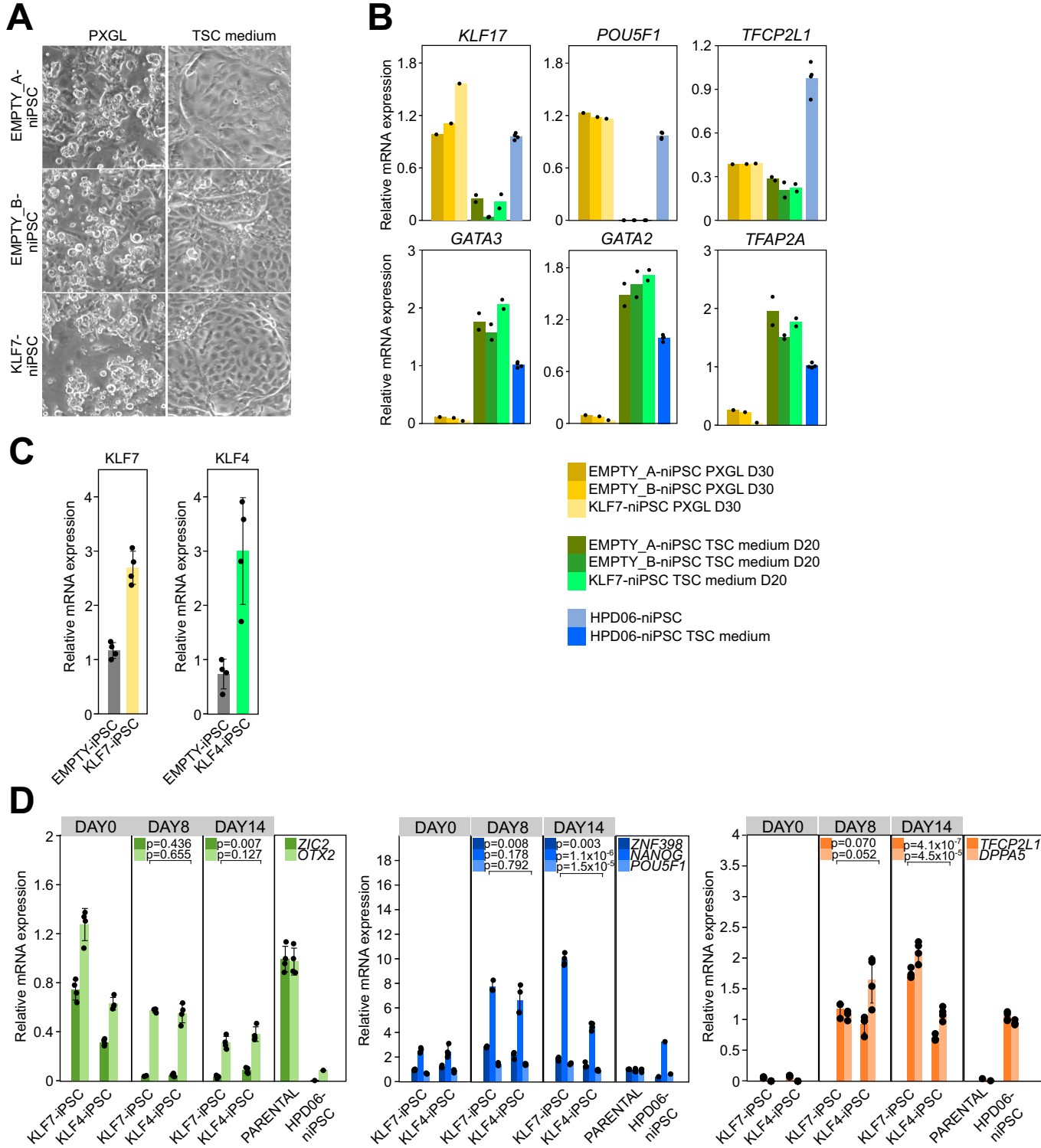

◀ **Figure EV3.   Testing KLF7 and KLF4 role in the induction of niPSCs.**

(A) Representative brightfield images naive iPSCs (niPSC) obtained by chemical resetting of two independent kiPSCs expressing an empty vector (EMPTY_A-niPSC and EMPTY_B-niPSC) and of kiPSCs expressing KLF7 (KLF7-niPSC), and cultured in PXGL naive medium or in TSC medium for 20 days. Representative images of 2 biological replicates are shown. Scale bar: 100 μm. (B) Expression by qPCR of naive (KLF17, POU5F1 and TFCP2L1) and trophoblast (GATA3, GATA2 and TFAP2A) markers in the indicated naive iPSC lines cultured in TSC medium for 20 days, bars indicate the mean of 2 biological replicates. HPD06 naive iPSCs, HDP06 TSCs and EMPTY_A/B- or KLF7-niPSCs were used as controls. Bars indicate the mean of 1 or 4 biological replicates, shown as dots. (C) Barplot showing expression by qPCR of KLF7 and KLF4 in kiPSCs transfected with a plasmid carrying the KLF7 or the KLF4 transgenes respectively. Mean +/− SD of 4 biological samples, shown as dots. (D) Expression by qPCR of conventional, shared and naive markers in kiPSC overexpressing an empty vector (EMPTY-iPSCs), the KLF7 transgene (KLF7-iPSCs) or the KLF4 transgene (KLF4-iPSCs) at day 0 and after 8 and 14 days of chemical resetting. Bars are the mean +/− SD of 4 biological samples, shown as dots. Parental kiPSCs and naive iPSCs HPD06 were used as controls, $n = 1$ or 4 biological replicates, shown as dots. Source data are available online for this figure.

**A**

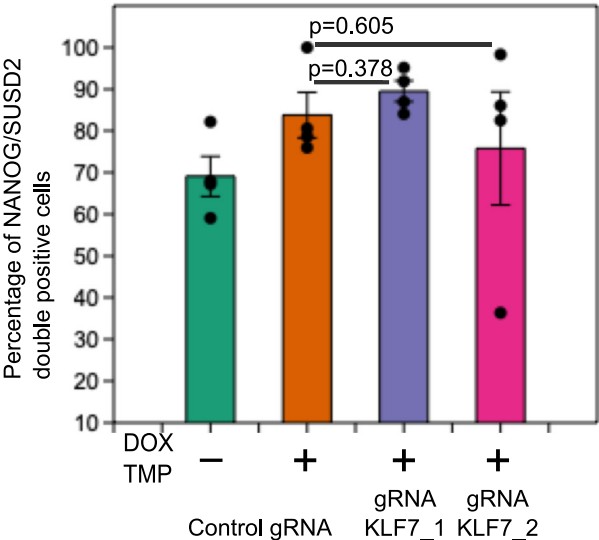

**Figure EV4.   Effect of KLF7 inhibition during chemical resetting.**

(**A**) Quantification of the percentage of cells expressing both NANOG and SUSD2 after 14 days of chemical resetting with control and KLF7-CRISPRi cells. See Fig. 4F for representative images. Bars indicate means of 4 biological replicates. Unpaired two-tailed *t* test. Source data are available online for this figure.

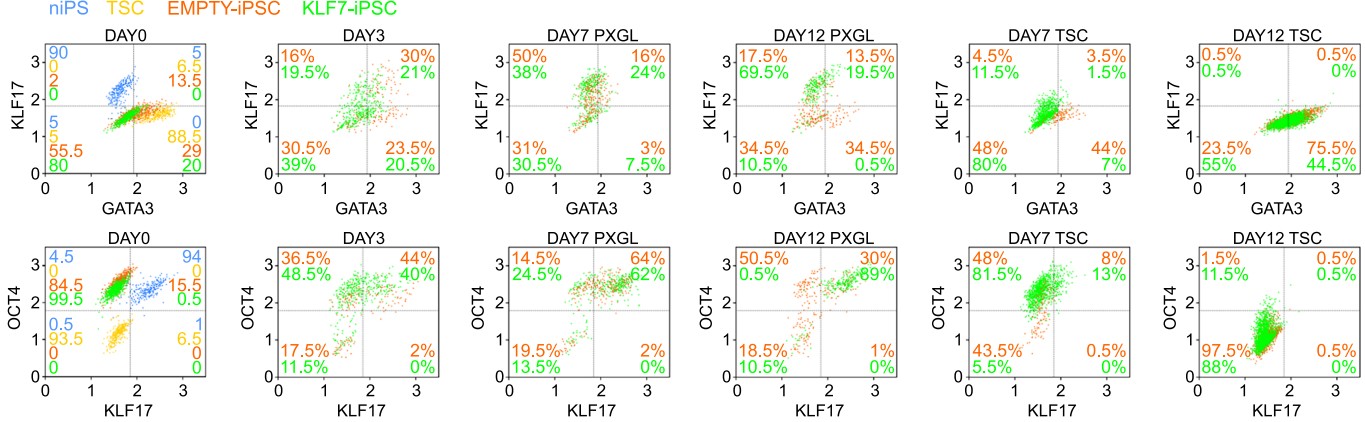

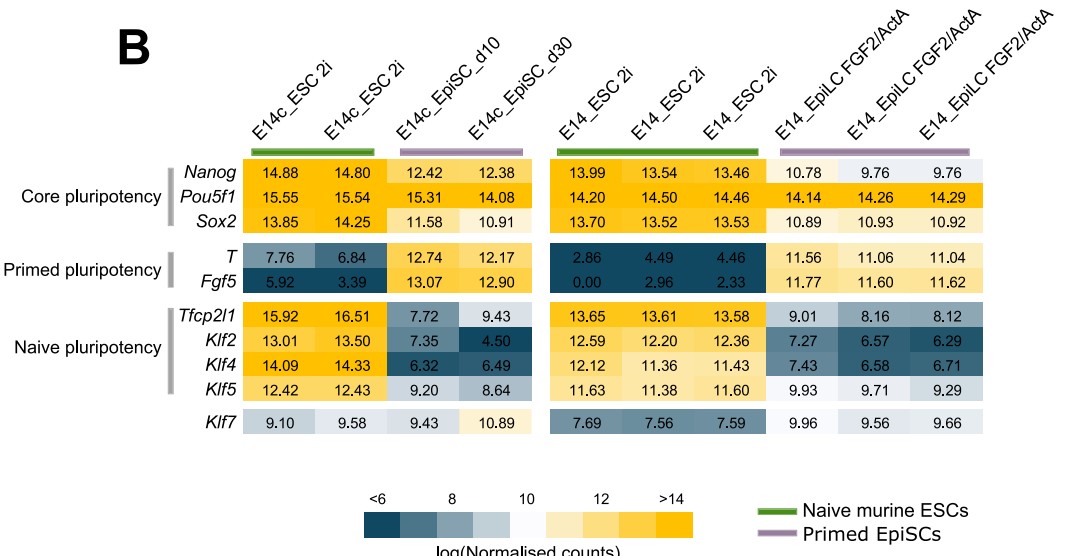

**Figure EV5.  Effect of KLF7 on chemical resetting and Klf7 expression analysis in murine PSCs.**

(A) Scatter plots of quantification of immunofluorescence signals for OCT4, KLF17 and GATA3. Log10 Integrated Intensity signal obtained from Cell Profile software analysis is displayed (See Methods). The percentage of cells in each quadrant is the mean of 2 independent experiments. (B) Heatmap showing the expression levels of core, naive and primed pluripotency markers in mouse naive ESCs, primed EpiSCs and EpiLCs. Data from (Fan et al, 2020; Zhang et al, 2016). Source data are available online for this figure.

