## [Peer Review File · EMBO Reports]

KLF7 is a general inducer of human pluripotency

Mattia Arboit, Irene Zorzan, Eleonora Pensabene, Marco Pellegrini, Federica Bertelli, Giorgia Panebianco, Davide Benvegnù, Giada Rossignoli, Paolo Martini, Gianluca Amadei, Elena Carbognin, and Graziano Martello

Corresponding author(s): Graziano Martello (graziano.martello@unipd.it), Elena Carbognin (elena.carbognin@unipd.it)

Review Timeline:

Submission Date:	19th Dec 23
Editorial Decision:	22nd Dec 23
Revision Received:	10th Jul 25
Editorial Decision:	22nd Aug 25
Revision Received:	16th Sep 25
Accepted:	26th Sep 25

Editor: Achim Breiling / Esther Schnapp

**Transaction Report: This manuscript was transferred to
EMBO reports following peer review at Review Commons.**

**Review
COMMONS**

Revision Plan

Manuscript number: RC-2023-02182

Corresponding author(s): Elena, Carbognin; Graziano, Martello

1. General Statements [optional]

This section is optional. Insert here any general statements you wish to make about the goal of the study or about the reviews

We thank the reviewers for their comments which have been extremely helpful, as well as the Review Commons editor for processing the manuscript.

We are pleased that the reviewers appreciated the significance of our work. Two important issues were raised by more than one reviewer, which we plan to address. One point was about the role of KLF7 during induction of pluripotency as compared to other Kruppel-like factors. To study this aspect, we plan to generate conventional/primed hPSCs with expression of KLF4 in order to assess the efficiency of chemical resetting compared to hPSCs with overexpression of KLF7. A second point raised was about the requirement of KLF7 for the maintenance of primed/conventional PSCs and during resetting. To this aim, we plan to knockdown KLF7 and study the biological characteristics of conventional/primed hPSCs and the effect on the derivation of naïve hPSCs.

Finally, we provide a point-by-point response to all reviewers' comments in the sections below.

2. Description of the planned revisions

Insert here a point-by-point reply that explains what revisions, additional experimentations and analyses are planned to address the points raised by the referees.

Reviewer #1 (Evidence, reproducibility and clarity (Required)):

Evidence reproducibility and clarity

The authors report that the human-specific KLF factor KLF7 can induce pluripotency in humans and can improve the reset toward naïve pluripotency when cells are cultured the PXGL medium. KLF7 falls behind KLF4 in reprogramming efficiency but might have a unique role in naïve reset (10-20 fold less efficient in iPSC colony yield). The topic of the study is interesting and adds important insights into the roles of KLF factors along the pluripotency continuum and pinpoints differences between mice and human. There are implications for stem cell engineering and boosting the developmental potency of stem cells (blastoid formation potential, interspecies chimera formation). However, some of the claims as to the unique role of KLF7 are unconvincing in the absence of comparison with other KLF factors, especially the Yamanaka factor KLF4. The flow and coherence of the text can be improved - at times reasoning and motivation of experiments are hard to follow

Major comments

- Why would a pan-pluripotency factor KLF7 which is expressed in both primed and naïve cells

Revision Plan

more potently trigger the naïve reset than the naïve specific factors KLF4/5/17? Such a comparison could widen the scope and interest of their work.. I would find it interesting if authors would compare the ability of key KLF factors to induce naivety. This is of particular interest as the overexpression of engineered Sox along with KLF4 was reported to improve the quality and developmental potential of PSC in multiple species (MacCarthy et al bioarxiv). Such an analysis could reveal unique features of KLF family members and lead to advanced stem cell models. They actually claim the SK naïve reset does not require naïve medium but the expression of SK alone is sufficient to induce this state. What do the authors think about this claim? Overall I feel the potential role of KLF7 in naïve reset is interesting but underdeveloped.

We thank the reviewer for the useful comments.

It has been shown in murine PSCs, that the pluripotency factors Nanog and Oct4 are expressed both at the naive and primed state and their forced expression, in combination with a medium supporting naive pluripotency, efficiently resets primed murine PSCs to naive (Radzishenskaya A. et al., 2013; Theunissen T. et al., 2011). It is therefore not surprising that a similar regulation might also be conserved in human and that the general pluripotency factor KLF7 is expressed in both states and drives efficient resetting.

Moreover, we agree that a direct comparison with another KLF factor could improve our work, so thanks to the reviewer's suggestions, we will generate conventional/primed hPSCs with exogenous KLF4 expression in order to assess the efficiency of chemical resetting compared to hPSCs with overexpression of KLF7.

Minor comments

- P3, line 52: "Surprisingly, however, KLF4 is also routinely used to generate conventional human iPSCs." Why is this surprising? KLF4 (and SOX2) are the most potent iPSC factors whilst MYC and OCT4 can be omitted (at least in mouse).

Thank you for pointing this out. We have rephrased the text accordingly (line 51).

- It would be nice if the demonstration of pluripotency and quality of KLF7 iPSC go beyond transcriptome profiling and included some further assays common in the field.

We assessed the quality of our OSK7M iPSCs by performing an EBs differentiation assay (Fig. 3d). We rephrased the text to further highlight this experiment (line 106). Of note, *in vivo* assays like teratoma formation are not allowed in Italy due to official regulations on animal testing.

- Fig 1A-B: color coding (of dots) is very confusing- which ones are PSCs and which ones are iPSCs? Another colour palette might fix. What is meant by "interrogating previously published data" (line 67)? Are these public RNA-seq data that were re-analyzed? I

We will highlight in the figures which cells are PSCs or iPSCs using different colours and shapes.

We rephrase the text to clarify that available RNA-seq data were reanalysed (line 67).

Revision Plan

- Fig 2b: how were the colony numbers obtained? By morphology, or using live cell staining? So form of staining is recommended colony counting (i.e. TRA-1-60).

We scored colonies both based on their morphology and after OCT4/NANOG staining. Actually, we observed that the counting based on morphology underestimated the number of iPSC colonies, so it is a more stringent method to score reprogrammed cells.

- Fig 2e: Also, they say that "[t]hree technical replicates were carried out for all quantitative PCR". Unless I'm mistaken, it seems that only two technical replicates were performed for these qPCR reactions (two dots visible per bar).

In figure 2e dots refer to two independent experiments. In each experiment we carried out three technical replicates for each sample.

- Fig 3c: "colture"; change to "culture" (and the title: "bone fide" should be "bona fide")

Thank you. We amended the typos in the figure and in the text (line 111).

- For Fig 2/3: since the paper is on KLF4/7, I'm surprised that expression levels of OCT4 and SOX2 were analysed but not KLF4. Given that the main finding was that KLF4 was not upregulated in PSCs, I would be interested to see what the KLF4 levels are like in the iPSCs. RNA-seq analysis/qPCR would be best; but if the authors would like to use other methods, that's fine too.

This is a good suggestion, we will add to Figure 3b the KLF4 expression levels.

- Fig 4: The explanatory text is too sparse. Readers should be reminded of the differences between of naïve and primed PSCs and the known roles of KLF4 (this could also be improved in the introduction). List names of naïve media used on top of author names (5iLA, PXGL, EPSCM etc). Why was HENSM by Hanna excluded?

We will amend the text explaining the main differences between naive and primed PSCs and the role of KLF4.

We will add PSCs derived in the HENSM medium in the analyses shown in figure 4.

- Fig 5: KLF7 is classified as a general pluripotency marker, but KLF4/KLF17 are classified as naïve markers. In that case, wouldn't it make more sense to overexpress a naïve specific marker in order to achieve naïve iPSCs at least as a control? What was the motivation here? I think the authors need to provide a more compelling reasoning why only KLF7 was studied or add more data for other KLFs (especially since it seems that the reprogramming efficiency of KLF4 is higher than that of KLF7 for conventional reprogramming (see Fig 2B)...)

We will perform resetting experiments using KLF4, as suggested, in order to compare the efficiency of KLF7 to a known naive factor.

o Fig 5B: the text currently says that the cells on the left side of Fig 5B are from Day7; but it says the cells are from Day0 in the actual figure. Which one is it? Also, based on how the text is written, do the cells on the left also contain EOS, or are they the wild-type variety?

Revision Plan

We agree that the text was confusing. Colonies appeared at day 7, but we showed them at day 12, when they were larger and easier to see. We amended the text accordingly. Moreover, the images at day 0 are simply the cell lines at the beginning of the resetting, which also contain EOS, as quantified on the right panels of Fig. 5b.

o Fig 5c: not all markers in this figure are naïve markers (as stated in the text); would suggest separating the markers and labelling them accordingly AND rewriting the text to reflect that.

We labelled the markers in the Fig. 5c as suggested by the reviewer and rephrased the text (line 136-137).

o Life cell reporters for naivety (CD75,SUSD2) could enrich this study.

We believe that the combination of bulk RNAseq and immunostaining for functional regulators of naive pluripotency (i.e. KLF17 and OCT4 (Lea et al., 2021 Development; Theunissen et al., 2014 Cell Stem Cell) are sufficient to described the acquisition of naive pluripotency.

- Schemes in 5A/6A could indicate when transgenes were added

For our chemical resetting experiments we used conventional hiPSCs (KiPS) with stable expression of KLF7 or an EMPTY vector (lines 126-127). We have also added this detail in the figure legends (line 291).

- Fig 7: the claim regard mouse pluripotency is a little outside of the scope of this paper; would recommend de-emphasizing the claim .

We will streamline the discussion and put less emphasis on murine PSCs.

- Could authors comment on the molecular features and whether there might be any non-redundant biochemical of KLF7 compared to other stemness-related KLFs? Looking at the conservation of the amino acids mediating base readout (-1,2,3,6) I expect specificity for DNA to be identical between KLF7 and KLF4 i.e. Figure S1A as reference for the C2H2 numbering convention: <https://www.cell.com/cms/10.1016/j.stemcr.2018.07.002/attachment/51171b7f-e644-4b0e-93c9-837632fd5d10/mmc1.pdf>

We thank the reviewer for the good suggestion that will be included in the revised manuscript.

- Similarly, are there features outside the DBD that might suggest a unique activity (IDR, TAD,PTM)? It seems KLF7 generates iPSCs much less efficiently than KLF4. Given the high similarity between their DBDs I wonder why this is so.

As above, this is an excellent point for discussion that will be added to revised manuscript.

Reviewer #1 (Significance (Required)):

Significance

• General assessment: The strength of the study is that the authors provide a potentially new way for the naïve reset in humans. This could improve human stem cell and embryo models. A

Revision Plan

limitation is that evidence is solely based on molecular (not functional) profiling and the uniqueness of KLF7 versus other KLF's (first and foremost KLF4) was not established.

- Advance: Findings on the human-specific role of KLF7 are novel and interesting especially the ability to facilitate the naïve reset. Yet, in the absence of a more systematic comparison with other methods (and KLF factors), the claim that KLF7 is essential for this feat is unconvincing.
- Audience: It's of interest to basic researchers in the broader stem cell community and those interested in early embryo development.

I work on cellular reprogramming, sequence-structure-function analysis of reprogramming factors and pluripotency.

Reviewer #2 (Evidence, reproducibility and clarity (Required)):

The naïve pluripotency is established in the inner cell mass (ICM) of blastocysts. After implantation, the naïve epiblast becomes primed for lineage specification. Pluripotent stem cells (PSCs) have been successfully derived from early embryos at different stages. In mice, stem cell derivations from ICM yield naïve ESCs. Primed PSCs derived from E5.5-7.5 epiblast are epiblast stem cells (EpiSCs). In humans, stem cell derivations from human embryos have yielded PSCs with features distinct from mouse ESCs and more like EpiSCs. Recently, naïve human PSCs have been directly isolated from pre-implantation epiblast or transformed from primed PSCs. Derivation of naïve hPSCs contributes to studying the molecular events of early lineage specification and accelerates the development of the generation of humanized organs in animal models from naïve hPSCs, opening an exciting avenue for regenerative medicine.

In this manuscript, the authors found that OSK7M could enable the reprogramming of human primary somatic cells. KLF7 is highly expressed in naïve PSCs and its forced expression in conventional hPSCs induces upregulation of naïve markers and boosts the efficiency of chemical resetting to naïve PSCs, suggesting that KLF7 is a general human pluripotency factor and an inducer of pluripotency. The new findings extend KLF7 function in naïve PSC generation and also provide references for the efficient generation of naïve PSCs. The people who focus on studying pluripotency and early embryo development might be interested in and influenced by the findings. The data are in general convincing. However, there are some issues that need to be resolved and improved.

Major comments:

1. Line 90: The authors showed that colonies derived from OSKM and OSK7M cocktails could be readily propagated for at least 10 passages. How many passages can OSK7M-iPSCs maintain in vitro prolonged culture?

And how about the pluripotency and developmental potential of OSK7M-iPSCs for a long-time culture? For example, pluripotency gene expression and teratoma formation.

We culture OSK7M-iPSCs up to 10 passages without noticing any abnormalities in the morphologies and duplication rate. However, we could extend such cultures for 5-10 more

Revision Plan

passages (i.e. a total of 2 months from iPSC generation) and perform staining for pluripotency markers or molecular analyses (by qPCR) and EBs differentiation assay to assess their developmental potential.

In vivo assays like teratoma formation are not allowed in Italy due to official regulations on animal testing.

2. Overexpression of KLF7 promotes the derivation of naïve PSCs. Are they different from naïve PSCs derived only by chemical resetting? For example, the pluripotency, the *in vitro* or *in vivo* developmental potential, and the efficiency of human blastoid generation.

A key feature of naïve PSCs is the potential to differentiate towards the trophoblast lineage in addition to the 3 germ layers. We will perform *in vitro* differentiation and EB formation assay to gauge the effect of KLF7 on differentiation potential.

However, establishing a human blastoid generation protocol would be beyond the scope of the current study.

As the manuscript mentioned, KLF7 is a general human pluripotency factor and an inducer of pluripotency. How does KLF7 knock-out affect the biological characteristics of hESCs? And whether KLF7 KO affects the derivation of naïve PSCs?

We agree that it would be informative to study the requirement of KLF7 for the maintenance of primed pluripotency and during resetting. We plan to do so either by knockdown or CRISPRi, depending on which technique allows efficient and controllable depletion of KLF7. It might be the case that a straight KO of KLF7 induces the collapse of primed PSCs, making resetting experiments not feasible.

3. Can naïve PSCs be directly reprogrammed from somatic cells with OSK7M under the PXGL medium? If so, how is the efficiency?

We believe that studying the role of KLF7 in the context of direct reprogramming of somatic cells to naïve pluripotency would go beyond the scope of this manuscript, as it would require substantial work for optimisation and generation of reagents.

Moreover, we think that both by over-expression and inhibition of KLF7 during resetting, we will be able to investigate its involvement in naïve pluripotency acquisition.

4. Figure 6d: The data showed that in PXGL medium, KiPS (EMPTY) contained about 66% of KLF17+ cells on day 7 and declined to 30% of KLF17+ cells on day 12. Why do KLF17+ cells (naïve PSCs) decline in PXGL medium?

Cells overexpressing KLF7 contained about 62% of KLF17+ cells on day 7 and increased to 89% of KLF17+ cells on day 12. Whether KLF7 function at this stage?

The reviewer raised an intriguing point, concerning the maintenance of naïve markers during resetting. Chemical resetting seems to induce transiently >60% of KLF17+/OCT4+ positive cells by day 7, however only a fraction of these cells is stabilised until day 12 (30%). In the presence of KLF7 overexpression, we observed a similar induction at day 7, which is maintained, or increased, up to day 12.

Revision Plan

This would indicate that KLF7 is important for the maintenance of a population of naive cells, rather than only for their induction.

We will add this important point to the discussion.

5. Figure 6e: The authors showed transcriptome analysis of KiPS KLF7 cells compared to KiPS16 EMPTY cells in standard culture conditions and found that trophoblast markers were not significantly changed. How is the gene expression during primed to naive transition or TSC differentiation?

We have already investigated this aspect, showing that at day 12 during primed to naive transition there is a strong induction of TSC markers, which is ablated by KLF7 expression (Fig. 5d). Quantitative immunostaining for GATA3 (TSC marker) confirmed this lack of activation in the presence of KLF7 (Fig. 6c).

Minor comments:

1. KLF7 is expressed in both primed and naive PSCs and when overexpressed in conventional PSCs, it enhances chemical resetting to naive PSCs. During primed to naive transition, how does the KLF7 gene expression pattern change?

This is a good suggestion, we will analyse the expression pattern of KLF7 during resetting.

2. Line 52: The reference should be added.

Thank you, we will add the reference.

3. Line 210-212: The reference should be added.

Thank you, we will add the reference.

Reviewer #2 (Significance (Required)):

The people who focus on studying pluripotency and early embryo development might be interested in and influenced by the findings.

The data are in general convincing. However, there are some issues that need to be resolved and improved.

Reviewer #3 (Evidence, reproducibility and clarity (Required)):

Summary:

In this manuscript, the authors found that KLF7 is generally expressed in both prime and naive human pluripotent stem cells. They showed that KLF7 could replace KLF4 to induce human iPSC cells in the microfluidic reprogramming system. The authors then found that overexpression of KLF7 in human prime iPSCs can facilitate the generation of naive iPSC cells. They also showed that KLF7 is a repressor of trophoblast markers. Collectively, these findings indicated that KLF7 is a general pluripotency inducer for human iPSC and naive iPSC induction.

Revision Plan

Major comments:

1. In Figure 2, as the reprogramming efficiency of OSK7M is much lower than that of OSKM, the authors should provide an OSM control to show whether the cells can be reprogrammed without KLF4 and KLF7.

We have performed the requested experiment (reprogramming with OSM only) as part of a manuscript in preparation. We observed an efficiency of reprogramming significantly lower than OSK7M, yet primary iPS colonies could be obtained.

We believe that this is due to the expression of KLF4 and KLF7 in human fibroblasts, as shown in Figure 4a.

2. It will be more convincing to perform a teratoma assay of OSK7M-iPSCs to demonstrate their multilineage differentiation potential.

In vivo assays like teratoma formation cannot be performed in Italy due to official regulations on animal testing.

However, we could extend such cultures for 5-10 more passages (i.e. a total of 2 months from iPSC generation) and perform staining for pluripotency markers or molecular analyses (by qPCR) and EBs differentiation assay to assess their multilineage differentiation potential.

3. Since KLF7 is also expressed in primed human iPS cells, the authors should show the expression level of KLF7 in the established KLF7-iPSC and EMPTY-iPS.

Good suggestion, we will add it to Figure 3b.

Minor comments:

The author claimed that KLF7 is a direct repressor of trophoblast markers, but the data in the manuscript cannot support this conclusion. The author can only claim that KLF7 can inhibit the expression of trophoblast markers.

We agree with the reviewer, and we believe that there was a misunderstanding. On pages 8-9 line 182-190 we also concluded that KLF7 regulates naive pluripotency markers, rather than trophoblast markers. We will rephrase the text to make it clearer.

Reviewer #3 (Significance (Required)):

KLF family proteins such as KLF4 and KLF17 have been identified as pluripotent inducers. In this study, the authors demonstrated that KLF7 is a novel pluripotent inducer of human IPS and naïve iPS cells, providing new insights into the functions of KLF family proteins in human pluripotency induction.

Revision Plan

3. Description of the revisions that have already been incorporated in the transferred manuscript

Please insert a point-by-point reply describing the revisions that were already carried out and included in the transferred manuscript. If no revisions have been carried out yet, please leave this section empty.

We have already carried out some revisions to the text and the figures, as suggested by Reviewer #1:

- P3, line 52: "Surprisingly, however, KLF4 is also routinely used to generate conventional human iPSCs." Why is this surprising? KLF4 (and SOX2) are the most potent iPSC factors whilst MYC and OCT4 can be omitted (at least in mouse).

Thank you for pointing this out. We have rephrased the text accordingly (line 51, highlighted in yellow).

- It would be nice if the demonstration of pluripotency and quality of KLF7 iPSC go beyond transcriptome profiling and included some further assays common in the field.

We assessed the quality of our OSK7M iPSCs by performing an EBs differentiation assay (Fig. 3d). We rephrased the text to further highlight this experiment (line 106, highlighted in yellow). Of note, *in vivo* assays like teratoma formation are not allowed in Italy due to official regulations on animal testing.

- Fig 1A-B: What is meant by "interrogating previously published data" (line 67)? Are these public RNA-seq data that were re-analyzed? I

We rephrase the text to clarify that available RNA-seq data were reanalysed (line 67, highlighted in yellow).

- Fig 3c: "colture"; change to "culture" (and the title: "bone fide" should be "bona fide")

Thank you. We amended the typos in Figure 3c and in the text (line 111, highlighted in yellow).

o Fig 5c: not all markers in this figure are naïve markers (as stated in the text); would suggest separating the markers and labelling them accordingly AND rewriting the text to reflect that.

We labelled the markers in the Fig. 5c as suggested by the reviewer and rephrased the text (line 136-137, highlighted in yellow).

- Schemes in 5A/6A could indicate when transgenes were added

For our chemical resetting experiments we used conventional hiPSCs (KiPS) with stable expression of KLF7 or an EMPTY vector (lines 126-127, highlighted in yellow). We have also added this detail in the figure legends (line 291, highlighted in yellow).

4. Description of analyses that authors prefer not to carry out

Please include a point-by-point response explaining why some of the requested data or additional analyses might not be necessary or cannot be provided within the scope of a revision. This can be due to time or resource limitations or in case of disagreement about the necessity of such additional data given the scope of the study. Please leave empty if not applicable.

Reviewer #1

- It would be nice if the demonstration of pluripotency and quality of KLF7 iPSC go beyond transcriptome profiling and included some further assays common in the field.

We assessed the quality of our OSK7M iPSCs by performing an EBs differentiation assay (Fig. 3d). We rephrased the text to further highlight this experiment (line 106, highlighted in yellow). Of note, *in vivo* assays like teratoma formation are not allowed in Italy due to official regulations on animal testing.

Reviewer #2

1. Line 90:

And how about the pluripotency and developmental potential of OSK7M-iPSCs for a long-time culture? For example, pluripotency gene expression and teratoma formation.

We could extend iPSCs cultures for 5-10 more passages (i.e. a total of 2 months from iPSC generation) and perform staining for pluripotency markers or molecular analyses (by qPCR) and EBs differentiation assay to assess their developmental potential.

However, we cannot perform *in vivo* assays like teratoma formation due to Italian regulations on animal testing.

2. Overexpression of KLF7 promotes the derivation of naïve PSCs. Are they different from naïve PSCs derived only by chemical resetting? For example, the pluripotency, the *in vitro* or *in vivo* developmental potential, and the efficiency of human blastoid generation.

A key feature of naïve PSCs is the potential to differentiate towards the trophoblast lineage in addition to the 3 germ layers. We will perform *in vitro* differentiation and EB formation assay to gauge the effect of KLF7 on differentiation potential.

However, establishing a human blastoid generation protocol, that will be time and technical demanding, would be beyond the scope of the current study.

3. Can naïve PSCs be directly reprogrammed from somatic cells with OSK7M under the PXGL medium? If so, how is the efficiency?

We believe that studying the role of KLF7 in the context of direct reprogramming of somatic cells to naïve pluripotency would go beyond the scope of this manuscript, as it would require substantial work for optimisation and generation of reagents.

Revision Plan

Moreover, we think that both by over-expression and inhibition of KLF7 during resetting, we will be able to investigate its involvement in naive pluripotency acquisition.

Reviewer #3

2. It will be more convincing to perform a teratoma assay of OSK7M-iPSCs to demonstrate their multilineage differentiation potential.

In vivo assays like teratoma formation cannot be performed in Italy due to official regulations on animal testing.

However, we could extend such cultures for 5-10 more passages (i.e. a total of 2 months from iPSC generation) and perform staining for pluripotency markers or molecular analyses (by qPCR) and EBs differentiation assay to assess their multilineage differentiation potential.

Dear Dr. Carbognin,

Thank you for the transfer of your research manuscript from Review Commons to EMBO reports. I now went through your manuscript, the referee reports from Review Commons (attached again below) and your revision plan. The referees have several comments, concerns, and suggestions to improve the manuscript, indicating that a major revision of the manuscript is necessary to allow publication of the study.

Going through your revision plan, it seems that most of these points will be adequately addressed during revision. I thus invite you to revise your manuscript accordingly with the understanding that all concerns must be addressed in the revised manuscript and/or in a detailed point-by-point response (as indicated in your revision plan). Acceptance of your manuscript will depend on a positive outcome of another round of review using the same set of referees. It is EMBO reports policy to allow a single round of major revision only and acceptance of the manuscript will therefore depend on the completeness of your responses included in the next, final version of the manuscript.

1) a .docx formatted version of the final manuscript text (including legends for main figures, EV figures and tables), but without the figures included. Figure legends should be compiled at the end of the manuscript text.

I think your manuscript could be published in the 'Report' format. For a Scientific Report we require that results and discussion sections are combined in a single chapter called "Results & Discussion". Please do this for your manuscript. For more details, please refer to our guide to authors:

<http://www.embopress.org/page/journal/14693178/authorguide#researcharticleguide>

Please note the limit of 25,000 (+/- 2,000) characters, excluding references and materials and methods.

2) individual production quality figure files as .eps, .tif, .jpg (one file per figure), of main figures (up to 5) and EV figures (up to 5) for a Report. Please upload these as separate, individual files upon re-submission.

4) a complete author checklist, which you can download from our author guidelines

(<https://www.embopress.org/page/journal/14693178/authorguide>). Please insert page numbers in the checklist to indicate where the requested information can be found in the manuscript. The completed author checklist will also be part of the RPF.

Please also follow our guidelines for the use of living organisms, and the respective reporting guidelines:
<http://www.embopress.org/page/journal/14693178/authorguide#livingorganisms>

5) that primary datasets produced in this study (e.g. RNA-seq, ChIP-seq, structural and array data) are deposited in an appropriate public database. If no primary datasets have been deposited, please also state this in a dedicated section (e.g. 'No primary datasets have been generated and deposited'), see below.

The accession numbers and database should be listed in a formal "Data Availability" section (placed after Materials & Methods) that follows the model below. This is now mandatory (like the COI statement). Please note that the Data Availability Section is restricted to new primary data that are part of this study. This section is mandatory. As indicated above, if no primary datasets have been deposited, please state this in this section

Data availability

8) Regarding data quantification and statistics, please make sure that the number "n" for how many independent experiments were performed, their nature (biological versus technical replicates), the bars and error bars (e.g. SEM, SD) and the test used to calculate p-values is indicated in the respective figure legends (also for potential EV figures and all those in the final Appendix). Please also check that all the p-values are explained in the legend, and that these fit to those shown in the figure. Please provide statistical testing where applicable. Please avoid the phrase 'independent experiment', but clearly state if these were biological or technical replicates. Please also indicate (e.g. with n.s.) if testing was performed, but the differences are not significant. In case n=2, please show the data as separate datapoints without error bars and statistics. See also: <http://www.embopress.org/page/journal/14693178/authorguide#statisticalanalysis>

9) Please also note our reference format:
<http://www.embopress.org/page/journal/14693178/authorguide#referencesformat>

10) We updated our journal's competing interests policy in January 2022 and request authors to consider both actual and perceived competing interests. Please review the policy <https://www.embopress.org/competing-interests> and update your competing interests if necessary. Please name this section 'Disclosure and Competing Interests Statement' and put it after the Acknowledgements section.

11) We now use CRediT to specify the contributions of each author in the journal submission system. CRediT replaces the author contribution section. Please use the free text box to provide more detailed descriptions and do not provide an author contributions section in the main manuscript text file. See also guide to authors:

12) Please add scale bars of similar style and thickness to all the microscopic images, using clearly visible black or white bars (depending on the background). Please place these in the lower right corner of the images themselves. Please do not write on or near the bars in the image but define the size in the respective figure legend.

13) Please add up to 5 keywords to the title page (below the abstract).

14) Please order the manuscript sections like this, using these names:

Title page - Abstract - Keywords - Introduction - Results & Discussion - Materials and Methods - Data availability section - Acknowledgements - Disclosure and Competing Interests Statement - References - Figure legends - Expanded View Figure legends

I look forward to seeing a revised version of your manuscript when it is ready. Please let me know if you have questions or comments regarding the revision.

Best,

Achim Breiling
Senior editor
EMBO reports

Referee #1:

The authors report that the human-specific KLF factor KLF7 can induce pluripotency in humans and can improve the reset toward naïve pluripotency when cells are cultured the PXGL medium. KLF7 falls behind KLF4 in reprogramming efficiency but might have a unique role in naïve reset (10-20-fold less efficient in iPSC colony yield). The topic of the study is interesting and adds important insights into the roles of KLF factors along the pluripotency continuum and pinpoints differences between mice and human. There are implications for stem cell engineering and boosting the developmental potency of stem cells (blastoid formation potential, interspecies chimera formation). However, some of the claims as to the unique role of KLF7 are unconvincing in the absence of comparison with other KLF factors, especially the Yamanaka factor KLF4. The flow and coherence of the text can be improved - at times reasoning and motivation of experiments are hard to follow

****Major comments****

- Why would a pan-pluripotency factor KLF7 which is expressed in both primed and naïve cells more potently trigger the naïve reset than the naïve specific factors KLF4/5/17? Such a comparison could widen the scope and interest of their work. I would find it interesting if authors would compare the ability of key KLF factors to induce naivety. This is of particular interest as the overexpression of engineered Sox along with KLF4 was reported to improve the quality and developmental potential of PSC in multiple species (MacCarthy et al bioarxiv). Such an analysis could reveal unique features of KLF family members and lead to advanced stem cell models. They actually claim the SK naïve reset does not require naïve medium but the expression of SK alone is sufficient to induce this state. What do the authors think about this claim? Overall I feel the potential role of KLF7 in naïve reset is interesting but underdeveloped.

****Minor comments****

- P3, line 52: "Surprisingly, however, KLF4 is also routinely used to generate conventional human iPSCs." Why is this surprising? KLF4 (and SOX2) are the most potent iPSC factors whilst MYC and OCT4 can be omitted (at least in mouse).
- It would be nice if the demonstration of pluripotency and quality of KLF7 iPSC go beyond transcriptome profiling and included some further assays common in the field.
- Fig 1A-B: color coding (of dots) is very confusing- which ones are PSCs and which ones are iPSCs? Another colour palette might fix What is meant by "interrogating previously published data" (line 67)? Are these public RNA-seq data that were re-analyzed? I
- Fig 2b: how were the colony numbers obtained? By morphology, or using live cell staining? So form of staining is recommended colony counting (i.e. TRA-1-60).
- Fig 2e: Also, they say that "[t]hree technical replicates were carried out for all quantitative PCR". Unless I'm mistaken, it seems that only two technical replicates were performed for these qPCR reactions (two dots visible per bar).
- Fig 3c: "colture"; change to "culture" (and the title: "bone fide" should be "bona fide")
- For Fig 2/3: since the paper is on KLF4/7, I'm surprised that expression levels of OCT4 and SOX2 were analysed but not KLF4. Given that the main finding was that KLF4 was not upregulated in PSCs, I would be interested to see what the KLF4 levels are like in the iPSCs. RNA-seq analysis/qPCR would be best; but if the authors would like to use other methods, that's fine too.

- Fig 4: The explanatory text is too sparse. Readers should be reminded of the differences between naïve and primed PSCs and the known roles of KLF4 (this could also be improved in the introduction). List names of naïve media used on top of author names (5iLA, PXGL, EPSCM etc). Why was HENSM by Hanna excluded?
- Fig 5: KLF7 is classified as a general pluripotency marker, but KLF4/KLF17 are classified as naïve markers. In that case, wouldn't it make more sense to overexpress a naïve specific marker in order to achieve naïve iPSCs at least as a control? What was the motivation here? I think the authors need to provide a more compelling reasoning why only KLF7 was studied or add more data for other KLFs (especially since it seems that the reprogramming efficiency of KLF4 is higher than that of KLF7 for conventional reprogramming (see Fig 2B)...)
 - Fig 5B: the text currently says that the cells on the left side of Fig 5B are from Day7; but it says the cells are from Day0 in the actual figure. Which one is it? Also, based on how the text is written, do the cells on the left also contain EOS, or are they the wild-type variety?
 - Fig 5c: not all markers in this figure are naïve markers (as stated in the text); would suggest separating the markers and labelling them accordingly AND rewriting the text to reflect that.
 - Life cell reporters for naivety (CD75,SUSD2) could enrich this study.
 - Schemes in 5A/6A could indicate when transgenes were added
 - Fig 7: the claim regard mouse pluripotency is a little outside of the scope of this paper; would recommend de-emphasizing the claim .
 - Could authors comment on the molecular features and whether there might be any non-redundant biochemical of KLF7 compared to other stemness-related KLFs? Looking at the conservation of the amino acids mediating base readout (-1,2,3,6) I expect specificity for DNA to be identical between KLF7 and KLF4 i.e. Figure S1A as reference for the C2H2numbering convention:
<https://www.cell.com/cms/10.1016/j.stemcr.2018.07.002/attachment/51171b7f-e644-4b0e-93c9-837632fd5d10/mmc1.pdf>
 - Similarly, are there features outside the DBD that might suggest a unique activity (IDR, TAD,PTM)? It seems KLF7 generates iPSCs much less efficiently than KLF4. Given the high similarity between their DBDs I wonder why this is so.

****Significance****

- General assessment: The strength of the study is that the authors provide a potentially new way for the naïve reset in humans. This could improve human stem cell and embryo models. A limitation is that evidence is solely based on molecular (not functional) profiling and the uniqueness of KLF7 versus other KLF's (first and foremost KLF4) was not established.
- Advance: Findings on the human-specific role of KLF7 are novel and interesting especially the ability to facilitate the naïve reset. Yet, in the absence of a more systematic comparison with other methods (and KLF factors), the claim that KLF7 is essential for this feat is unconvincing.
- Audience: It's of interest to basic researchers in the broader stem cell community and those interested in early embryo development.

 Referee #2:

Naïve pluripotency is established in the inner cell mass (ICM) of blastocysts. After implantation, the naïve epiblast becomes primed for lineage specification. Pluripotent stem cells (PSCs) have been successfully derived from early embryos at different stages. In mice, stem cell derivations from ICM yield naïve ESCs. Primed PSCs derived from E5.5-7.5 epiblast are epiblast stem cells (EpiSCs). In humans, stem cell derivations from human embryos have yielded PSCs with features distinct from mouse ESCs and more like EpiSCs. Recently, naïve human PSCs have been directly isolated from pre-implantation epiblast or transformed from primed PSCs. Derivation of naïve hPSCs contributes to studying the molecular events of early lineage specification and accelerates the development of the generation of humanized organs in animal models from naïve hPSCs, opening an exciting avenue for regenerative medicine.

In this manuscript, the authors found that OSK7M could enable the reprogramming of human primary somatic cells. KLF7 is highly expressed in naïve PSCs and its forced expression in conventional hPSCs induces upregulation of naïve markers and boosts the efficiency of chemical resetting to naïve PSCs, suggesting that KLF7 is a general human pluripotency factor and an inducer of pluripotency. The new findings extend KLF7 function in naïve PSC generation and also provide references for the efficient generation of naïve PSCs. The people who focus on studying pluripotency and early embryo development might be interested in and influenced by the findings. The data are in general convincing. However, there are some issues that need to be resolved and improved.

****Major comments:****

1. Line 90: The authors showed that colonies derived from OSKM and OSK7M cocktails could be readily propagated for at least 10 passages. How many passages can OSK7M-iPSCs maintain in vitro prolonged culture? And how about the pluripotency and developmental potential of OSK7M-iPSCs for a long-time culture? For example, pluripotency gene expression and teratoma formation.
2. Overexpression of KLF7 promotes the derivation of naïve PSCs. Are they different from naïve PSCs derived only by chemical resetting? For example, the pluripotency, the in vitro or in vivo developmental potential, and the efficiency of human blastoid

generation. As the manuscript mentioned, KLF7 is a general human pluripotency factor and an inducer of pluripotency. How does KLF7 knock-out affect the biological characteristics of hESCs? And whether KLF17 KO affects the derivation of naïve PSCs?

3. Can naïve PSCs be directly reprogrammed from somatic cells with OSK7M under the PXGL medium? If so, how is the efficiency?
4. Figure 6d: The data showed that in PXGL medium, KiPS (EMPTY) contained about 66% of KLF17+ cells on day 7 and declined to 30% of KLF17+ cells on day 12. Why do KLF17+ cells (naïve PSCs) decline in PXGL medium? Cells overexpressing KLF7 contained about 62% of KLF17+ cells on day 7 and increased to 89% of KLF17+ cells on day 12. Whether KLF7 function at this stage?
5. Figure 6e: The authors showed transcriptome analysis of KiPS KLF7 cells compared to KiPS16 EMPTY cells in standard culture conditions and found that trophoblast markers were not significantly changed. How is the gene expression during primed to naïve transition or TSC differentiation?

****Minor comments:****

1. KLF7 is expressed in both primed and naïve PSCs and when overexpressed in conventional PSCs, it enhances chemical resetting to naïve PSCs. During primed to naïve transition, how does the KLF7 gene expression pattern change?
2. Line 52: The reference should be added.
3. Line 210-212: The reference should be added.

****Significance****

People who focus on studying pluripotency and early embryo development might be interested in and influenced by the findings.

The data are in general convincing. However, there are some issues that need to be resolved and improved.

Referee #3:

In this manuscript, the authors found that KLF7 is generally expressed in both prime and naïve human pluripotent stem cells. They showed that KLF7 could replace KLF4 to induce human iPS cells in the microfluidic reprogramming system. The authors then found that overexpression of KLF7 in human prime iPSCs can facilitate the generation of naïve iPS cells. They also showed that KLF7 is a repressor of trophoblast markers. Collectively, these findings indicated that KLF7 is a general pluripotency inducer for human iPS and naïve iPS induction.

****Major comments:****

1. In Figure 2, as the reprogramming efficiency of OSK7M is much lower than that of OSKM, the authors should provide an OSM control to show whether the cells can be reprogrammed without KLF4 and KLF7.
2. It will be more convincing to perform a teratoma assay of OSK7M-iPSCs to demonstrate their multilineage differentiation potential.
3. Since KLF7 is also expressed in primed human iPS cells, the authors should show the expression level of KLF7 in the established KLF7-iPSC and EMPTY-iPS.

****Minor comments:****

The author claimed that KLF7 is a direct repressor of trophoblast markers, but the data in the manuscript cannot support this conclusion. The author can only claim that KLF7 can inhibit the expression of trophoblast markers.

****Significance****

KLF family proteins such as KLF4 and KLF17 have been identified as pluripotent inducers. In this study, the authors demonstrated that KLF7 is a novel pluripotent inducer of human iPS and naïve iPS cells, providing new insights into the functions of KLF family proteins in human pluripotency induction.

Full Revision

Manuscript number: RC-2023-02182 / EMBOR-2023-58682V2

Corresponding author(s): Elena, Carbognin ; Graziano, Martello

[Please use this template only if the submitted manuscript should be considered by the affiliate journal as a full revision in response to the points raised by the reviewers.]

*If you wish to submit a preliminary revision with a revision plan, please use our "Revision Plan" template. **It is important to use the appropriate template to clearly inform the editors of your intentions.***

1. General Statements [optional]

We thank the reviewers for their comments, which have been extremely helpful, as well as the Review Commons editor for processing our manuscript.

We are pleased that the reviewers appreciated the significance of our work. Two important issues were raised by more than one reviewer, and we fully addressed these. One point pertained to the role of KLF7 during induction of pluripotency as compared to other Kruppel-like factors. To study this aspect, we generated conventional/primed hPSCs with expression of KLF4 in order to assess the efficiency of chemical resetting compared to hPSCs with overexpression of KLF7. We now show that expression of KLF7 during chemical resetting resulted in the formation of a morphologically homogenous population of naïve colonies, whereas overexpression of KLF4 during resetting led to the formation of a more heterogeneous population. In addition, the expression of general pluripotency markers and of naïve markers was more pronounced when resetting with KLF7. A second point raised was about the requirement of KLF7 for the maintenance of primed/conventional PSCs and during resetting. To this aim, we knocked down KLF7 and studied the biological characteristics of conventional/primed hPSCs and the effect on the derivation of naïve hPSCs. We now show that while KLF7 knockdown does not affect the maintenance of conventional hPSCs, it greatly decreases the efficiency of resetting to naïve hPSCs, indicating that it is necessary for this process.

Below, we provide a point-by-point response to all reviewers' comments.

This section is mandatory. Please insert a point-by-point reply describing the revisions that were already carried out and included in the transferred manuscript.

Below is the response detailing how suggestions by reviewer 1 were incorporated in the text and figures. In the manuscript, changes to the text are highlighted in yellow.

Reviewer #1 (Evidence, reproducibility and clarity (Required)):

Evidence reproducibility and clarity

The authors report that the human-specific KLF factor KLF7 can induce pluripotency in humans and can improve the reset toward naïve pluripotency when cells are cultured the PXGL medium. KLF7 falls behind KLF4 in reprogramming efficiency but might have a unique role in naïve reset (10-20 fold less efficient in iPSC colony yield). The topic of the study is interesting and adds important insights into the roles of KLF factors along the pluripotency continuum and pinpoints differences between mice and human. There are implications for stem cell engineering and boosting the developmental potency of stem cells (blastoid formation potential, interspecies chimera formation). However, some of the claims as to the unique role of KLF7 are unconvincing in the absence of comparison with other KLF factors, especially the Yamanaka factor KLF4. The flow and coherence of the text can be improved - at times reasoning and motivation of experiments are hard to follow

Major comments

- Why would a pan-pluripotency factor KLF7 which is expressed in both primed and naïve cells more potently trigger the naïve reset than the naïve specific factors KLF4/5/17? Such a comparison could widen the scope and interest of their work.. I would find it interesting if authors would compare the ability of key KLF factors to induce naïvety. This is of particular interest as the overexpression of engineered Sox along with KLF4 was reported to improve the quality and developmental potential of PSC in multiple species (MacCarthy et al bioarxiv). Such an analysis could reveal unique features of KLF family members and lead to advanced stem cell models. They actually claim the SK naïve reset does not require naïve medium but the expression of SK alone is sufficient to induce this state. What do the authors think about this claim? Overall I feel the potential role of KLF7 in naïve reset is interesting but underdeveloped.

We thank the reviewer for the useful comments.

It has been shown in murine PSCs, that the pluripotency factor Oct4 is expressed both at the naïve and primed state and its forced expression, together with Nanog in combination with a medium supporting naïve pluripotency, efficiently resets primed murine PSCs to naïve (Radzishenskaya A. et al., 2013; Theunissen T. et al., 2011). It is therefore not surprising that a similar regulation might also be conserved for some human transcription factors involved in pluripotency regulation. This is even more evident with human NANOG which is expressed in both the naïve and primed embryonic stem cells, similarly to the general pluripotency factor KLF7, which is also expressed in both states and drives efficient resetting.

Moreover, we agree that a direct comparison with another KLF factor could improve our work, so thanks to the reviewer's suggestions, we have generated conventional/primed hPSCs with exogenous KLF4 expression in order to assess the efficiency of chemical resetting compared to hPSCs with overexpression of KLF7 (Fig. 5F). We observe that resetting conventional PSCs using KLF7 gave rise to a morphologically homogeneous population whereas using KLF4 resulted in a higher degree of heterogeneity. Furthermore, the general pluripotency markers and naïve markers showed higher expression levels in response to KLF7 overexpression during resetting than in response to KLF4 overexpression (lines 183-194, highlighted).

Full Revision

- P3, line 52: "Surprisingly, however, KLF4 is also routinely used to generate conventional human iPSCs." Why is this surprising? KLF4 (and SOX2) are the most potent iPSC factors whilst MYC and OCT4 can be omitted (at least in mouse).

Thank you for pointing this out. We have rephrased the text accordingly (line 52, highlighted in yellow).

- It would be nice if the demonstration of pluripotency and quality of KLF7 iPSC go beyond transcriptome profiling and included some further assays common in the field.

We assessed the quality of our OSK7M iPSCs by performing an EBs differentiation assay (Fig. 3d). We rephrased the text to further highlight this experiment (line 123, highlighted in yellow). Of note, *in vivo* assays like teratoma formation are not allowed in Italy due to official regulations on animal testing.

- Fig 1A-B: color coding (of dots) is very confusing- which ones are PSCs and which ones are iPSCs? Another colour palette might fix.

We have changed the colour palette of this graph to improve readability

- Fig 1A-B: What is meant by "interrogating previously published data" (line 67)? Are these public RNA-seq data that were re-analyzed? I

We rephrased the text to clarify that available RNA-seq data were reanalyzed (line 80, highlighted in yellow).

- Fig 2b: how were the colony numbers obtained? By morphology, or using live cell staining? So form of staining is recommended colony counting (i.e. TRA-1-60).

We scored colonies both based on their morphology and after OCT4/NANOG staining. Actually, we observed that the counting based on morphology underestimated the number of iPSC colonies, so it is a more stringent method to score reprogrammed cells.

- Fig 2e: Also, they say that "[t]hree technical replicates were carried out for all quantitative PCR". Unless I'm mistaken, it seems that only two technical replicates were performed for these qPCR reactions (two dots visible per bar).

In figure 2e dots refer to two independent experiments. In each experiment we carried out three technical replicates for each sample.

- Fig 3c: "colture"; change to "culture" (and the title: "bone fide" should be "bona fide")

Thank you. We amended the typos in Figure 3c and in the text (line 129, highlighted in yellow).

- For Fig 2/3: since the paper is on KLF4/7, I'm surprised that expression levels of OCT4 and SOX2 were analysed but not KLF4. Given that the main finding was that KLF4 was not upregulated in PSCs, I would be interested to see what the KLF4 levels are like in the iPSCs. RNA-seq analysis/qPCR would be best; but if the authors would like to use other methods, that's fine too.

This is a good suggestion, we have added to Figure 3b KLF4 expression levels.

- Fig 4: The explanatory text is too sparse. Readers should be reminded of the differences between of naïve and primed PSCs and the known roles of KLF4 (this could also be improved in the introduction). List names of naïve media used on top of author names (5iLA, PXGL, EPSCM etc). Why was HENSM by Hanna excluded?

We have amended the text explaining the main differences between naïve and primed PSCs (lines 39-49, highlighted) and the role of KLF4 (lines 50-54, highlighted).

We have added PSCs derived in the HENSM medium in the analyses shown in figure 4.

- Fig 5: KLF7 is classified as a general pluripotency marker, but KLF4/KLF17 are classified as naïve markers. In that case, wouldn't it make more sense to overexpress a naïve specific marker in order to achieve naïve iPSCs at least as a control? What was the motivation here? I think the authors need to provide a more compelling reasoning why only KLF7 was studied or add more data for other KLFs (especially since it seems that the reprogramming efficiency of KLF4 is higher than that of KLF7 for conventional reprogramming (see Fig 2B)...).

As mentioned before, expression of a transcription factor during both the naïve and primed pluripotency developmental stage does not preclude the possibility that it may have an important role in one stage or the other. Our rationale for focusing specifically on KLF7 was its very strong expression at these stages compared to the other Klf factors, strongly suggesting that it may be involved in human development.

We have performed resetting experiments using KLF4, as suggested, in order to compare the efficiency of KLF7 to a known naïve factor (Fig. 5F). We observed that resetting conventional PSCs using KLF7 gave rise to a morphologically homogeneous population whereas using KLF4 resulted in a higher degree of heterogeneity. Furthermore, the general pluripotency markers and naïve markers showed higher expression levels in response to KLF7 overexpression during resetting (lines 183-194, highlighted).

o Fig 5B: the text currently says that the cells on the left side of Fig 5B are from Day7; but it says the cells are from Day0 in the actual figure. Which one is it? Also, based on how the text is written, do the cells on the left also contain EOS, or are they the wild-type variety?

We agree that the text was confusing. Colonies appeared at day 7, but we showed them at day 12, when they were larger and easier to see. We amended the text of the figure legend

Full Revision

accordingly. Moreover, the images at day 0 are simply the cell lines at the beginning of the resetting, which also contain EOS, as quantified on the right panels of Fig. 5b.

o Fig 5c: not all markers in this figure are naïve markers (as stated in the text); would suggest separating the markers and labelling them accordingly AND rewriting the text to reflect that.

We labelled the markers in the Fig. 5c as suggested by the reviewer and rephrased the text (line 161 , highlighted in yellow).

- Schemes in 5A/6A could indicate when transgenes were added

For our chemical resetting experiments we used conventional hiPSCs (KiPS) with stable expression of KLF7 or an EMPTY vector (lines 149-151, highlighted in yellow). We have also added this detail in the figure legends (line 385, highlighted in yellow).

- Fig 7: the claim regard mouse pluripotency is a little outside of the scope of this paper; would recommend de-emphasizing the claim .

We have streamlined the discussion and put less emphasis on murine PSCs.

- Could authors comment on the molecular features and whether there might be any non-redundant biochemical of KLF7 compared to other stemness-related KLFs? Looking at the conservation of the amino acids mediating base readout (-1,2,3,6) I expect specificity for DNA to be identical between KLF7 and KLF4 i.e. Figure S1A as reference for the C2H2 numbering convention: <https://www.cell.com/cms/10.1016/j.stemcr.2018.07.002/attachment/51171b7f-e644-4b0e-93c9-837632fd5d10/mmc1.pdf>

We thank the reviewer for this good suggestion, which is now included in the discussion of our revised manuscript.

- Similarly, are there features outside the DBD that might suggest a unique activity (IDR, TAD,PTM)? It seems KLF7 generates iPSCs much less efficiently than KLF4. Given the high similarity between their DBDs I wonder why this is so.

As above, this is an excellent point, which we have also added to the discussion of our revised manuscript.

- It would be nice if the demonstration of pluripotency and quality of KLF7 iPSC go beyond transcriptome profiling and included some further assays common in the field.

We assessed the quality of our OSK7M iPSCs by performing an EBs differentiation assay (Fig. 3d). We rephrased the text to further highlight this experiment (line 123, highlighted in yellow). Of

Full Revision

note, *in vivo* assays like teratoma formation are not allowed in Italy due to official regulations on animal testing.

- Life cell reporters for naivety (CD75,SUSD2) could enrich this study.

We believe that the combination of bulk RNAseq and immunostaining for functional regulators of naive pluripotency (i.e. KLF17 and OCT4 (Lea et al., 2021 Development; Theunissen et al., 2014 Cell Stem Cell) are sufficient to describe the acquisition of naive pluripotency.

Reviewer #1 (Significance (Required)):

Significance

- General assessment: The strength of the study is that the authors provide a potentially new way for the naïve reset in humans. This could improve human stem cell and embryo models. A limitation is that evidence is solely based on molecular (not functional) profiling and the uniqueness of KLF7 versus other KLF's (first and foremost KLF4) was not established.
- Advance: Findings on the human-specific role of KLF7 are novel and interesting especially the ability to facilitate the naïve reset. Yet, in the absence of a more systematic comparison with other methods (and KLF factors), the claim that KLF7 is essential for this feat is unconvincing.
- Audience: It's of interest to basic researchers in the broader stem cell community and those interested in early embryo development.

I work on cellular reprogramming, sequence-structure-function analysis of reprogramming factors and pluripotency.

Here is the response detailing how suggestions by reviewer #2 were incorporated in the text and figures. In the manuscript, changes to the text are highlighted in yellow.

Reviewer #2 (Evidence, reproducibility and clarity (Required)):

The naïve pluripotency is established in the inner cell mass (ICM) of blastocysts. After implantation, the naïve epiblast becomes primed for lineage specification. Pluripotent stem cells (PSCs) have been successfully derived from early embryos at different stages. In mice, stem cell derivations from ICM yield naïve ESCs. Primed PSCs derived from E5.5-7.5 epiblast are epiblast stem cells (EpiSCs). In humans, stem cell derivations from human embryos have yielded PSCs with features distinct from mouse ESCs and more like EpiSCs. Recently, naïve human PSCs have been directly isolated from pre-implantation epiblast or transformed from primed PSCs. Derivation of naïve hPSCs contributes to studying the molecular events of early lineage specification and accelerates the development of the generation of humanized organs in animal models from naïve hPSCs, opening an exciting avenue for regenerative medicine.

In this manuscript, the authors found that OSK7M could enable the reprogramming of human primary somatic cells. KLF7 is highly expressed in naïve PSCs and its forced expression in

conventional hPSCs induces upregulation of naive markers and boosts the efficiency of chemical resetting to naive PSCs, suggesting that KLF7 is a general human pluripotency factor and an inducer of pluripotency. The new findings extend KLF7 function in naive PSC generation and also provide references for the efficient generation of naive PSCs. The people who focus on studying pluripotency and early embryo development might be interested in and influenced by the findings. The data are in general convincing. However, there are some issues that need to be resolved and improved.

Major comments:

1. Line 90: The authors showed that colonies derived from OSKM and OSK7M cocktails could be readily propagated for at least 10 passages. How many passages can OSK7M-iPSCs maintain in vitro prolonged culture?

And how about the pluripotency and developmental potential of OSK7M-iPSCs for a long-time culture? For example, pluripotency gene expression and teratoma formation.

We cultured OSK7M-iPSCs for up to 10 passages without noticing any abnormalities in the morphologies and duplication rate. However, we extended such cultures for 5-10 more passages (i.e. a total of 2 months from iPSC generation) and performed staining for pluripotency markers or molecular analyses (by qPCR) and EBs differentiation assay to assess their developmental potential (see for instance Fig. 5D). We find that the morphology, proliferation and differentiation potential of our OSK7M-iPSCs were not affected at later passages.

In vivo assays like teratoma formation are not allowed in Italy due to official regulations on animal testing.

2. Overexpression of KLF7 promotes the derivation of naive PSCs. Are they different from naive PSCs derived only by chemical resetting? For example, the pluripotency, the in vitro or in vivo developmental potential, and the efficiency of human blastoid generation.

A key feature of naive PSCs is the potential to form extraembryonic cell types, like Trophoblast Stem Cells (TSCs). We thus performed TSC differentiation on naive iPSCs generated by KLF7 expression (KLF7-niPSCs) (Fig. EV2A,B). We observed rapid shut down of pluripotency markers and upregulation of TSC markers, to levels comparable to control niPSCs. We conclude that KLF7 expression during resetting did not affect pluripotency shut down and extraembryonic potential. We respectfully chose not to test the developmental potential of these cells by blastoid formation because establishing a human blastoid generation protocol would be technically very demanding, since it is not a model we use, and we believe it would be beyond the scope of the current study.

As the manuscript mentioned, KLF7 is a general human pluripotency factor and an inducer of pluripotency. How does KLF7 knock-out affect the biological characteristics of hESCs? And whether KLF7 KO affects the derivation of naive PSCs?

We agree that it would be informative to study the requirement of KLF7 for the maintenance of primed pluripotency and during resetting. We have downregulated KLF7 by CRISPRi (Fig. 6). and we now show that KLF7 depletion does not affect long term maintenance and pluripotency of established conventional PSCs but it greatly decreases the number of naïve colonies obtained upon chemical resetting. We conclude that KLF7 is not necessary for maintenance of conventional PSCs but it greatly promotes the efficiency of chemical resetting.

3. Can naïve PSCs be directly reprogrammed from somatic cells with OSK7M under the PXGL medium? If so, how is the efficiency?

We believe that studying the role of KLF7 in the context of direct reprogramming of somatic cells to naïve pluripotency would go beyond the scope of this manuscript, as it would require substantial work for optimization and generation of reagents.

We have investigated KLF7 involvement in naïve pluripotency acquisition by both overexpression and inhibition of KLF7 during resetting.

4. Figure 6d: The data showed that in PXGL medium, KiPS (EMPTY) contained about 66% of KLF17+ cells on day 7 and declined to 30% of KLF17+ cells on day 12. Why do KLF17+ cells (naïve PSCs) decline in PXGL medium?

Cells overexpressing KLF7 contained about 62% of KLF17+ cells on day 7 and increased to 89% of KLF17+ cells on day 12. Whether KLF7 function at this stage?

The reviewer raised an intriguing point, concerning the maintenance of naïve markers during resetting. Chemical resetting seems to induce transiently >60% of KLF17+/OCT4+ positive cells by day 7, however only a fraction of these cells is stabilized until day 12 (30%). In the presence of KLF7 overexpression, we observed a similar induction at day 7, which is maintained, or increased, up to day 12.

This would indicate that KLF7 is important for the maintenance of a population of naïve cells, rather than only for their induction.

We have added this important point to the discussion.

5. Figure 6e: The authors showed transcriptome analysis of KiPS KLF7 cells compared to KiPS16 EMPTY cells in standard culture conditions and found that trophoblast markers were not significantly changed. How is the gene expression during primed to naïve transition or TSC differentiation?

We have already investigated this aspect, showing that at day 12 during primed to naïve transition in PXGL medium we found that 34.5% of cells expressed trophoblast transcription factor GATA3, however this fraction of cells was reduced to 0.5% in cells expressing KLF7. (Fig. 5d). This result was also confirmed by quantitative immunostaining for GATA3 (TSC marker) (Fig. 6c).

Minor comments:

Full Revision

1. KLF7 is expressed in both primed and naive PSCs and when overexpressed in conventional PSCs, it enhances chemical resetting to naive PSCs. During primed to naive transition, how does the KLF7 gene expression pattern change?

We thank the reviewer for this great suggestion. We have analyzed the expression pattern of KLF7 during resetting and, very interestingly, we observe a peak in KLF7 expression at day 3 of resetting (Fig. 4C), suggesting that KLF7 is involved in this process. Consistently with this interpretation, KLF7 downregulation during chemical resetting greatly decreases its efficiency (Fig. 6C-G).

2. Line 52: The reference should be added.

Thank you, we have added the relevant reference (line 61 of the revised manuscript).

3. Line 210-212: The reference should be added.

Thank you, we have added the relevant reference (line 289 of the revised manuscript).

Reviewer #2 (Significance (Required)):

The people who focus on studying pluripotency and early embryo development might be interested in and influenced by the findings.

The data are in general convincing. However, there are some issues that need to be resolved and improved.

Here is the response detailing how suggestions by reviewer #3 were incorporated in the text and figures. In the manuscript, changes to the text are highlighted in yellow.

Reviewer #3 (Evidence, reproducibility and clarity (Required)):

Summary:

In this manuscript, the authors found that KLF7 is generally expressed in both prime and naive human pluripotent stem cells. They showed that KLF7 could replace KLF4 to induce human iPS cells in the microfluidic reprogramming system. The authors then found that overexpression of KLF7 in human prime iPSCs can facilitate the generation of naive iPS cells. They also showed that KLF7 is a repressor of trophoblast markers. Collectively, these findings indicated that KLF7 is a general pluripotency inducer for human iPS and naive iPS induction.

Major comments:

1. In Figure 2, as the reprogramming efficiency of OSK7M is much lower than that of OSKM, the authors should provide an OSM control to show whether the cells can be reprogrammed without KLF4 and KLF7.

Full Revision

We have performed the requested experiment (reprogramming with OSM only) as part of a manuscript in preparation. We observed an efficiency of reprogramming significantly lower than OSK7M, yet primary iPS colonies could be obtained.

We believe that this is due to the expression of KLF4 and KLF7 in human fibroblasts, as shown in Figure 4a.

2. It will be more convincing to perform a teratoma assay of OSK7M-iPSCs to demonstrate their multilineage differentiation potential.

In vivo assays like teratoma formation cannot be performed in Italy due to official regulations on animal testing.

However, we could extend such cultures for 5-10 more passages (i.e. a total of 2 months from iPSC generation) and perform staining for pluripotency markers or molecular analyses (by qPCR) and EBs differentiation assay to assess their multilineage differentiation potential.

3. Since KLF7 is also expressed in primed human iPS cells, the authors should show the expression level of KLF7 in the established KLF7-iPSC and EMPTY-iPS.

We thank the reviewer for the good suggestion, we have added this to Figure 3b.

Minor comments:

The author claimed that KLF7 is a direct repressor of trophoblast markers, but the data in the manuscript cannot support this conclusion. The author can only claim that KLF7 can inhibit the expression of trophoblast markers.

We agree with the reviewer, and we believe that there was a misunderstanding. On page 12 line 249-251 we also concluded that KLF7 regulates naive pluripotency markers, rather than trophoblast markers. We have rephrased the text to make it clearer.

Reviewer #3 (Significance (Required)):

KLF family proteins such as KLF4 and KLF17 have been identified as pluripotent inducers. In this study, the authors demonstrated that KLF7 is a novel pluripotent inducer of human IPS and naïve iPS cells, providing new insights into the functions of KLF family proteins in human pluripotency induction.

Dear Dr. Martello,

Thank you for the submission of your revised manuscript to our editorial offices. I have now received the reports from the two referees that I asked to re-evaluate the study, you will find below. Original referee #3 was completely unresponsive to my invitations to re-assess the study. But going through your point-by-point-response, I consider his/her points as adequately addressed.

As you will see, referees #1 and #2 indicate that the manuscript has improved and many of their points have been addressed. However, both referees have remaining concerns or further comments and suggestions to improve the manuscript, I ask you to address in a final revised manuscript. Please also provide a final p-b-p-response to these points and my editorial requests below.

Editorial requests:

- We plan to publish your manuscript as report. For a scientific report we need 5 final main figures and up to 5 final EV figures. Please arrange your figures in a way to have not more than 5 main figures and up to 5 EV figures. Please then update their legends and all the affected callouts accordingly. Present Figure 8 can be used as synopsis image (see below) and removed.

- Please add up to 5 Keywords to the manuscript and order the sections like this, using only these names:
Title page - Abstract - Keywords - Introduction - Results & Discussion - Methods - Data availability section - Acknowledgements (please put here all the funding information) - Disclosure and Competing Interests Statement - References - Figure legends - Expanded View Figure legends

- We now use CRediT to specify the contributions of each author in the journal submission system. CRediT replaces the author contribution section. Please use the free text box to provide more detailed descriptions and do NOT provide your final manuscript text file with an author contributions section. See also our guide to authors:
<https://www.embopress.org/page/journal/14693178/authorguide#authorshipguidelines>

- Some of the scale bars in the microscopic images are rather thin and hard to see. Please improve.

- Please check again that the number "n" for how many independent experiments were performed, their nature (biological versus technical replicates), the bars and error bars (e.g. SEM, SD) and the test used to calculate p-values is indicated in the respective figure legends. Please also check that all the p-values are explained in the legend, and that these fit to those shown in the figure. Please provide statistical testing where applicable. Please avoid the phrase 'independent experiment' but clearly state if these were biological or technical replicates. Please also indicate (e.g. with n.s.) if testing was performed, but the differences are not significant. In case n=2, please show the data as separate datapoints without error bars and statistics. See also:
<http://www.embopress.org/page/journal/14693178/authorguide#statisticalanalysis>

If n<5, please show single datapoints for diagrams. Moreover:

- Please note that information related to n is missing in the legend of figure 7E
- Please note that the error bars are not defined in the legend of figure EV1 C
- Please indicate the statistical test used for data analysis in the legend of figure EV1 C

- Please add to each legend (main, EV figures and Appendix Figures, where applicable) a 'Data Information' section explaining the statistics used or providing information regarding replicates and scales. See:

- - Please remove now the referee tokens from the data availability section and make sure that all deposited datasets are public latest upon online publication of the manuscript. The data availability section is restricted to externally deposited large datasets generated in a study. Please remove all other information from this section.

- All Materials and Methods need to be described in the main text using our 'Structured Methods' format, which is required for all research articles. According to this format, the Methods section should include a Reagents and Tools Table (listing key reagents, experimental models, software, and relevant equipment and including their sources and relevant identifiers), uploaded as separate file, and a Methods section in which we encourage the authors to describe their methods using a step-by-step protocol format with bullet points, to facilitate the adoption of the methodologies across labs. More information on how to adhere to this format as well as downloadable templates (.doc) for the Reagents and Tools Table can be found in our author guidelines (section 'Structured Methods'):

Please add the information provided in Tables EV2-4 directly to the reagents and tools table. Please update any callouts and also provide further callouts to the reagents and tools table where appropriate.

- Table EV1 is a dataset. Please name this Dataset EV1 and upload it as dataset file with a legend on the first TAB. Please add callouts for this dataset to the main manuscript text file.

- Please make sure that all the funding information is also entered into the online submission system and that it is complete and similar to the one in the acknowledgement section of the manuscript text file. Presently PRIN2022 PNRR project (EMBRYODIET) is missing as separate funder in the submission system. Please check.

In addition, I would need from you uploaded separately:

I look forward to seeing the final revised version of your manuscript when it is ready.

Please let me know if you have questions regarding the revision.

Best,

Referee #1:

We have previously looked into this study for Review Commons, which is now being considered for EMBO Reports. The authors have prepared a detailed response and performed a number of additional experiments, which we appreciated. The KLF family has several members with unique and redundant functions in early development. Working out their individual roles and aspects of their evolution and biochemical uniqueness is relevant and interesting. I find the improvements and revisions helpful, and I am overall positive that the study is sufficiently advanced to eventually warrant the broad exposure through EMBO reports. A main concern is that authors constantly claim that KLF7 is a human-specific TF (which it is not).

Further, an outperformance over KLF4 in pushing the naïve reset is overstated, as there are no obvious quantitative data backing such a claim, just anecdotal evidence based on a few marker genes and morphological observations. We believe that KLF7 can replace KLF4 in this assay, but we are not convinced that it works any better.

Specific remarks:

1. Introduction: "In contrast, a human-specific KLF, named KLF7, is robustly expressed in conventional PSCs and supports pluripotency downstream of TGF-beta (Zorzan et al, 2020). "This statement is wrong. Please review the studies on KLF evolution, including <https://academic.oup.com/gbe/article/7/8/2289/557806>. I do not understand what the claim on human specificity is based on as KLF6/7 appear to be ancient factors. KLF4 might be a bilaterian-specific TF (KLFs are overall ancestral). A more precise discussion about the evolution of C2H2 zinc fingers in general and the KLFs groups in particular would be helpful.
2. Lines 87-88: The rationale for comparing KLF7 expression specifically to PRDM14 and ZNF398, rather than more commonly referenced naïve pluripotency markers, is unclear. Is there any specific reason? Suggest explaining in the main text.
3. Lines 94-96: The phrasing of the hypothesis, "...we hypothesised that expression of KLF7, instead of KLF4, together with OCT4, SOX2 and cMYC (OSK7M) could enable reprogramming..." is problematic. It implies KLF4 could not enable reprogramming, which is well-established to be false. The hypothesis should be rephrased to focus on the better performance or complementary role of KLF7.
4. Fig. 3A: The figure legend and main text lack critical details describing the iPSC lines used to generate the heatmap. Please specify: 1) The number of independent biological replicates (clonal lines) analyzed for each condition (OSKM vs. OSK7M); 2) Whether the data points represent technical replicates from a single clone or measurements from different clones.
5. Fig. 3D: 1) The germ layer origin (ectoderm, mesoderm, endoderm) for the markers shown in the panel should be clearly labeled; 2) suggest to include the expression levels of other core pluripotency markers (e.g., SOX2, NANOG) in the day 0 iPSC lines; 3) The schematic in Fig. 3C indicates an analysis at day 5, but corresponding qPCR data is not shown. Please either remove the day 5 reference from the schematic or provide the missing data; 4) Including phase-contrast images of the EBs and

ICC staining for the lineage markers would significantly strengthen the evidence for spontaneous differentiation.

6. Image Quality: Phase-contrast images are of low resolution and it is hard to appreciate cell morphology accurately. This issue is present in Figure 5B, 5F, 7B, E1A, and E2A.

7. Comparison of KLF7 and KLF4 Potency: The conclusion that KLF7 is "more potent" than KLF4 in chemical resetting is based on limited data (a few marker genes and morphology) and is not fully supported. To make this claim more convincingly, the authors should perform a more comprehensive comparison or tone down this part. If they insist on this claim, repeating the key analyses from Figure 7 (which likely detail a robust characterization of the naïve state) or human blastoid generation for KLF4-overexpressing cells would provide a direct and informative comparison to substantiate their conclusion.

8. Line 272: typo "KL7".

9. Discussion: The study demonstrates a modest increase in resetting efficiency with KLF7 but does not explore its functional significance in advanced models like human blastoid generation or any in vivo experiments of development potential. The discussion should be expanded to address the potential broader implications and advantages of using KLF7 for naïve resetting beyond mere efficiency. What are the hypothesized mechanistic benefits? How might this impact downstream applications?

10. Lines 309-310: The statement regarding KLF7 expression is incorrect and misrepresents the cited paper. Figure 5B from the study of Yamane et al., 2018 shows the expression level of Klf7 in mouse embryonic stem cells is indeed comparable to that of Klf2, Klf4, and Klf5. This claim must be corrected.

11. Lines 311-312 / Conclusion: The claim that "KLF7 is a human specific pluripotency regulator, whose function is not conserved in rodents" is overstated and unfair. As noted in comment #10, its expression in mouse cells is significant. Furthermore, while its role in mouse naïve pluripotency may not be fully defined, its robust expression has been documented (e.g., Zorzan et al., 2020; Yamane et al., 2018). The conclusion should be toned down to accurately reflect the data.

12. Page 54: Are KLF factors really all absent in primed mouse PSCs? Re-analysis of some public data and showing this (in the supplement) would help strengthen this point.

13. Does KLF7 (like many other pluripotency factors) have a role in human germ cell specification?

Referee #2:

In this manuscript, the authors found that KLF7 is generally expressed in both prime and naïve human pluripotent stem cells. OSK7M could enable the reprogramming of human primary somatic cells. They also showed that KLF7 is a repressor of trophoblast markers and overexpression of KLF7 in human prime iPSCs can facilitate the generation of naïve iPS cells.

The data are in general convincing. However, there are some issues that need to be resolved and improved.

1、 The cell images in Figure 2A, Figure 5B, Figure 5F and Figure7B are too blurry.

2、 During PNT, it is common to observe mosaic expression of marker genes like KLF17 within cell clones, where only a subset of cells is positive for markers. Therefore, using clone counts alone to compare transformation efficiency across different cell lines lacks rigor, as shown in Figure 6G.

3、 In the manuscript, the effect of KLF7 deletion in PNT was examined. Whether KLF7 deletion can affect the efficiency of somatic cell reprogramming into iPSCs and differentiation into TSCs remains unclear.

4、 The displayed method of the cell proportions in Figure 7D is prone to misinterpretation, and only two experiments replicates were conducted.

Dear Editor,

We are happy that the reviewers appreciated our effort towards addressing all of their suggestions and concerns. Below, we include a detailed point-by-point response.

Dear Dr. Martello,

Thank you for the submission of your revised manuscript to our editorial offices. I have now received the reports from the two referees that I asked to re-evaluate the study, you will find below. Original referee #3 was completely unresponsive to my invitations to re-assess the study. But going through your point-by-point-response, I consider his/her points as adequately addressed.

As you will see, referees #1 and #2 indicate that the manuscript has improved and many of their points have been addressed. However, both referees have remaining concerns or further comments and suggestions to improve the manuscript, I ask you to address in a final revised manuscript. Please also provide a final p-b-p-response to these points and my editorial requests below.

Editorial requests:

- We plan to publish your manuscript as report. For a scientific report we need 5 final main figures and up to 5 final EV figures. Please arrange your figures in a way to have not more than 5 main figures and up to 5 EV figures. Please then update their legends and all the affected callouts accordingly. Present Figure 8 can be used as synopsis image (see below) and removed.

We have now reformatted our manuscript to include five main figures and 5 EV figures; texts, legends and source data have all been updated accordingly. We provide figure 8 as a synopsis image in the required format.

- Please add up to 5 Keywords to the manuscript and order the sections like this, using only these names: Title page - Abstract - Keywords - Introduction - Results & Discussion - Methods - Data availability section - Acknowledgements (please put here all the funding information) - Disclosure and Competing Interests Statement - References - Figure legends - Expanded View Figure legends

We have formatted the manuscript according to the guidelines above and provided 5 keywords.

- We now use CRediT to specify the contributions of each author in the journal submission system. CRediT replaces the author contribution section. Please use the free text box to provide more detailed descriptions and do NOT provide your final manuscript text file with an author contributions section. See also our guide to authors:

We have removed author contributions from the manuscript and we have included them using CRediT

<https://www.embopress.org/page/journal/14693178/authorguide#authorshipguidelines>

- Some of the scale bars in the microscopic images are rather thin and hard to see. Please improve.

We have improved all the scale bars in the manuscript to increase their visibility

- Please check again that the number "n" for how many independent experiments were performed, their nature (biological versus technical replicates), the bars and error bars (e.g. SEM, SD) and the test used to calculate p-values is indicated in the respective figure legends. Please also check that all the p-values are explained in the legend, and that these fit to those shown in the figure. Please provide statistical testing where applicable. Please avoid the phrase 'independent experiment' but clearly state if these were biological or technical replicates. Please also indicate (e.g. with n.s.) if testing was performed, but the differences are not significant. In case n=2, please show the data as separate datapoints without error bars and statistics. See also:

<http://www.embopress.org/page/journal/14693178/authorguide#statisticalanalysis>

We have carefully checked the figure legends to ensure that all of the required information is provided. Importantly, we have clarified whether a replicate was technical or biological, but we would also like to point out that by independent experiment we refer to a biological set that was performed or collected on a different day from another biological set.

If $n < 5$, please show single datapoints for diagrams. Moreover:

- Please note that information related to n is missing in the legend of figure 7E
- Please note that the error bars are not defined in the legend of figure EV1 C
- Please indicate the statistical test used for data analysis in the legend of figure EV1 C

We have ensured that we now provide the required information

- Please add to each legend (main, EV figures and Appendix Figures, where applicable) a 'Data Information' section explaining the statistics used or providing information regarding replicates and scales. See:

This information for every panel in our manuscript can already be found in the corresponding section of the relevant figure legend. It is necessary to keep them separate because often in a figure there are different types of data and it is confusing to summarize the statistics in just one line.

- Please remove now the referee tokens from the data availability section and make sure that all deposited datasets are public latest upon online publication of the manuscript. The data availability section is restricted to externally deposited large datasets generated in a study. Please remove all other information from this section.

The tokens have been removed from the data availability section

- All Materials and Methods need to be described in the main text using our 'Structured Methods' format, which is required for all research articles. According to this format, the Methods section should include a Reagents and Tools Table (listing key reagents, experimental models, software, and relevant equipment and including their sources and relevant identifiers), uploaded as separate file, and a Methods section in which we encourage the authors to describe their methods using a step-by-step protocol format with bullet points, to facilitate the adoption of the methodologies across labs. More information on how to adhere to this format as well as downloadable templates (.doc) for the Reagents and Tools Table can be found in our author guidelines (section 'Structured Methods'):

We now provide the required Reagents and Tools Table.

- Please add the information provided in Tables EV2-4 directly to the reagents and tools table. Please update any callouts and also provide further callouts to the reagents and tools table where appropriate.

All these tables are now part of the Reagents and Tools table and all callouts have been updated accordingly.

- Table EV1 is a dataset. Please name this Dataset EV1 and upload it as dataset file with a legend on the first TAB. Please add callouts for this dataset to the main manuscript text file.

We now include Table EV1 as Dataset EV1 and all callouts have been updated accordingly.

- Please make sure that all the funding information is also entered into the online submission system and that it is complete and similar to the one in the acknowledgement section of the manuscript text file. Presently PRIN2022 PNRR project (EMBRYODIET) is missing as separate funder in the submission system. Please check.

We now provide all of the funding information in the online submission system and we ensured that it matches what is provided in the acknowledgement section of the manuscript.

In addition, I would need from you uploaded separately:

We now provide this additional information as separate files. The schematic / synopsis figure is provided in the format required.

I look forward to seeing the final revised version of your manuscript when it is ready.

Please let me know if you have questions regarding the revision.

Best,

Point-by-point response

Referee #1:

We have previously looked into this study for Review Commons, which is now being considered for EMBO Reports. The authors have prepared a detailed response and performed a number of additional experiments, which we appreciated. The KLF family has several members with unique and redundant functions in early development. Working out their individual roles and aspects of their evolution and biochemical uniqueness is relevant and interesting. I find the improvements and revisions helpful, and I am overall positive that the study is sufficiently advanced to eventually warrant the broad exposure through EMBO reports. A main concern is that authors constantly claim that KLF7 is a human-specific TF (which it is not).

Further, an outperformance over KLF4 in pushing the naïve reset is overstated, as there are no obvious quantitative data backing such a claim, just anecdotal evidence based on a few marker genes and morphological observations. We believe that KLF7 can replace KLF4 in this assay, but we are not convinced that it works any better.

Specific remarks:

1. Introduction: "In contrast, a human-specific KLF, named KLF7, is robustly expressed in conventional PSCs and supports pluripotency downstream of TGF-beta (Zorzan et al, 2020). "This statement is wrong. Please review the studies on KLF evolution, including <https://academic.oup.com/gbe/article/7/8/2289/557806>. I do not understand what the claim on human specificity is based on as KLF6/7 appear to be ancient factors. KLF4 might be a bilaterian-specific TF (KLFs are overall ancestral). A more precise discussion about the evolution of C2H2 zinc fingers in general and the KLFs groups in particular would be helpful.

We apologize for this misunderstanding; we can see now that our phrasing was misleading. When we wrote "human-specific", we were referring to KLF7 function in supporting chemical resetting and reprogramming specifically in the human model. In Zorzan et al. 2020, when we overexpressed KLF7 in mouse EpiSC we did not observe any difference. Nonetheless, to avoid further confusion, we have removed "human-specific" from the text and we have reworded it to explain this properly.

2. Lines 87-88: The rationale for comparing KLF7 expression specifically to PRDM14 and ZNF398, rather than more commonly referenced naïve pluripotency markers, is unclear. Is there any specific reason? Suggest explaining in the main text.

We apologize to this reviewer for not being sufficiently clear. It is true that in Fig. 1B we did not compare KLF7 expression against the more commonly referenced naïve pluripotency markers, such as POU5F1 or NANOG but that is simply because we already show them in Fig.1A and hence it is still possible to visually compare them even though they are not on the same graph. In addition to these, we also compared KLF7 expression to PRDM14 and ZNF398 because they are well-established as genes that are functionally relevant to pluripotency. For instance, in (Chia et al. 2010, Pubmed ID: 20953172), the authors clearly showed that PRDM14 played a role in the maintenance of hESC identity and in the reacquisition of pluripotency. In Zorzan et al. 2020, our own work showed that inhibition of ZNF398 during somatic cell reprogramming impeded activation of pluripotency and epithelial genes and inhibited colony formation. (lines 95-97).

3. Lines 94-96: The phrasing of the hypothesis, "...we hypothesised that expression of KLF7, instead of KLF4, together with OCT4, SOX2 and cMYC (OSK7M) could enable reprogramming..." is problematic. It implies KLF4 could not enable reprogramming, which is well-established to be false. The hypothesis should be rephrased to focus on the better performance or complementary role of KLF7.

We thank the reviewer for pointing this out. We have now changed that line to: "Since KLF7 sustains pluripotency in conventional PSCs (Zorzan *et al*, 2020), we hypothesized that expression of KLF7 could perform a similar function as KLF4 and, together with OCT4, SOX2 and cMYC (OSK7M), could enable reprogramming of primary somatic cells" (98-101).

4. Fig. 3A: The figure legend and main text lack critical details describing the iPSC lines used to generate the heatmap. Please specify: 1) The number of independent biological replicates (clonal lines) analyzed for each

condition (OSKM vs. OSK7M); 2) Whether the data points represent technical replicates from a single clone or measurements from different clones.

We apologize to this reviewer for not being sufficiently clear. Regarding point 1, we utilized two clones per condition and the clones were analyzed twice, at two different passages. Regarding point 2, the data points represent technical replicates from a single clone. We have now added this information to the figure legend.

5. Fig. 3D: 1) The germ layer origin (ectoderm, mesoderm, endoderm) for the markers shown in the panel should be clearly labeled; 2) suggest to include the expression levels of other core pluripotency markers (e.g., SOX2, NANOG) in the day 0 iPSC lines; 3) The schematic in Fig. 3C indicates an analysis at day 5, but corresponding qPCR data is not shown. Please either remove the day 5 reference from the schematic or provide the missing data; 4) Including phase-contrast images of the EBs and ICC staining for the lineage markers would significantly strengthen the evidence for spontaneous differentiation.

We thank this reviewer for pointing this out. Regarding point 1, we have now added the germ layer of origin to the markers. Regarding point 2, we already show the expression levels for SOX2 and NANOG, together with the expression of 10 more markers in Fig.3B, now the new (Fig. 1G). Regarding point 3, the data for day 5 is shown in Figure EV1B, now the new (Fig. EV2C). We have only analyzed the day 5 data for passage 7 because we observed that the transcriptional changes were still ongoing and became much more evident and conclusive at day 15, which was the only timepoint analyzed at the later passages. Regarding point 4, we now provide phase-contrast images of the Embryoid Bodies (Fig. EV2B) and explain in the text that "After 15 days we could not detect differences in the size and shape of EBs (Fig. EV2B)" (lines 134-135).

6. Image Quality: Phase-contrast images are of low resolution and it is hard to appreciate cell morphology accurately. This issue is present in Figure 5B, 5F, 7B, E1A, and E2A.

We apologize to this reviewer, the low quality of those panels was due to a compression error during export. We will make sure to provide high resolutions exports.

7. Comparison of KLF7 and KLF4 Potency: The conclusion that KLF7 is "more potent" than KLF4 in chemical resetting is based on limited data (a few marker genes and morphology) and is not fully supported. To make this claim more convincingly, the authors should perform a more comprehensive comparison or tone down this part. If they insist on this claim, repeating the key analyses from Figure 7 (which likely detail a robust characterization of the naïve state) or human blastoid generation for KLF4-overexpressing cells would provide a direct and informative comparison to substantiate their conclusion.

We agree with the reviewer that we should tone down this claim. We have now amended the abstract, results and discussion to say that KLF7 and KLF4 function nearly identically in our assays. While we report that we did observe some differences, we stress that more work is needed to uncover their functional significance.

8. Line 272: typo "KL7".

We thank the reviewer for pointing this out, we fixed this typo.

9. Discussion: The study demonstrates a modest increase in resetting efficiency with KLF7 but does not explore its functional significance in advanced models like human blastoid generation or any in vivo experiments of development potential. The discussion should be expanded to address the potential broader implications and advantages of using KLF7 for naïve resetting beyond mere efficiency. What are the hypothesized mechanistic benefits? How might this impact downstream applications?

We thank the reviewer for this comment. We have now expanded the relevant section of the discussion to speculate on the potential impact for downstream applications, such as blastoid formation: "It remains to be tested whether utilization of either KLF4 or KLF7 is preferable in specific assays, for instance

blastoid formation. Human blastoids have been generated with several approaches and utilizing human naive ESCs cultured in different conditions, however in all these protocols a very small number of blastoids develops to a post-implantation-like stage, and even then, some cell types present in human post-implantation embryos are missing. It would be interesting to test whether our naive ESCs reset with KLF7 could yield blastoids that develop with higher efficiency or can form additional post-implantation cell types.”

10. Lines 309-310: The statement regarding KLF7 expression is incorrect and misrepresents the cited paper. Figure 5B from the study of Yamane et al., 2018 shows the expression level of Klf7 in mouse embryonic stem cells is indeed comparable to that of Klf2, Klf4, and Klf5. This claim must be corrected.

We respectfully disagree that KLF7 expression level in mouse embryonic stem cells is comparable to Klf2, Klf4 and Klf5. In the study cited by the reviewer, Figure 5B shows that the expression level of KLF2 and KLF5 is approximately 10^{100} FPKM, and KLF4 is approximately 10^{50} , instead the expression of KLF7 is approx. 10^5 . In the KLF family, only KLF17, KLF14 and KLF1 show lower expression levels. To further confirm this, as suggested by this reviewer in point 12, we performed a new analysis of published datasets, showing that it independently validates the low expression of KLF7 in mouse ESCs and primed EpiSCs/EpiLCs. We show this analysis in Fig. EV5B and we discuss it in lines 305-315 of the text.

11. Lines 311-312 / Conclusion: The claim that "KLF7 is a human specific pluripotency regulator, whose function is not conserved in rodents" is overstated and unfair. As noted in comment #10, its expression in mouse cells is significant. Furthermore, while its role in mouse naive pluripotency may not be fully defined, its robust expression has been documented (e.g., Zorzan et al., 2020; Yamane et al., 2018). The conclusion should be toned down to accurately reflect the data.

We agree with this reviewer that our claim that KLF7 is human-specific is inaccurate and misleading, we meant to say that this function we describe in pluripotency is specific to human pluripotent stem cells. Importantly, this claim is based on Zorzan et al. 2020, in which we showed that KLF7 overexpression in 2 EpiSC lines had no effect. We have rephrased the text accordingly to clarify that: "Of note, KLF7 is not expressed at significant levels in murine PSCs, and its forced expression in primed EpiSCs is inconsequential (Zorzan *et al*, 2020), indicating that KLF7 function as a human pluripotency regulator, is not conserved in rodents.

12. Page 54: Are KLF factors really all absent in primed mouse PSCs? Re-analysis of some public data and showing this (in the supplement) would help strengthen this point.

We thank the reviewer for this comment and we apologize for not being sufficiently clear. We were referring exclusively to KLF2/4/5, not to all KLF factors. Nonetheless, we have performed the analysis suggested by this reviewer (See comment 10).

13. Does KLF7 (like many other pluripotency factors) have a role in human germ cell specification?

We thank this reviewer for the excellent question. We do not have an answer for that but we have added this point to the discussion: "It remains to be seen whether KLF7, similarly to other pluripotency factors, also plays a role in human germ cell specification" (Lines 326-327).

Referee #2:

In this manuscript, the authors found that KLF7 is generally expressed in both prime and naïve human pluripotent stem cells. OSK7M could enable the reprogramming of human primary somatic cells. They also showed that KLF7 is a repressor of trophoblast markers and overexpression of KLF7 in human prime iPSCs can facilitate the generation of naïve iPS cells.

The data are in general convincing. However, there are some issues that need to be resolved and improved.

1. The cell images in Figure 2A, Figure 5B, Figure 5F and Figure 7B are too blurry.

We thank this reviewer for pointing this out, we have now replaced those panels with high resolution images. This problem was due to a compression error during export.

2. During PNT, it is common to observe mosaic expression of marker genes like KLF17 within cell clones, where only a subset of cells is positive for markers. Therefore, using clone counts alone to compare transformation efficiency across different cell lines lacks rigor, as shown in Figure 6G.

We thank the reviewer for this comment. In Fig. EV4 we have quantified the percentage of Nanog/SUSD2 double-positive cells after 14 days of chemical resetting in control conditions and with CRISPRi against KLF7. We now show that the percentage of double positive cells does not significantly change across the different conditions, even though the number of clones is significantly decreased upon KLF7 downregulation (Fig. 4F,G). All together, this indicates that the majority of the clones that we counted comprised Nanog/SUSD2 double-positive cells. We have amended the main text to explain this (lines 218-225).

3. In the manuscript, the effect of KLF7 deletion in PNT was examined. Whether KLF7 deletion can affect the efficiency of somatic cell reprogramming into iPSCs and differentiation into TSCs remains unclear.

We thank the reviewer for this suggestion but we respectfully believe that it would be beyond the scope of this paper.

4. The displayed method of the cell proportions in Figure 7D is prone to misinterpretation, and only two experiments replicates were conducted.

We apologise to this reviewer for the confusion. We have now changed the way in which we represent these data to improve the clarity, the new panel can be found in Fig 5D. We have extracted the relevant populations from our previous analysis for this barplot. The bars show the mean and the points are the individual experiments. We have amended the text accordingly to be consistent with this new panel.

Graziano Martello
University of Padova
Department of Biology
Viale G. Colombo, 3
Padova 35131
Italy

Dear Graziano,

I am very pleased to accept your manuscript for publication in the next available issue of EMBO reports. Thank you for your contribution to our journal.
